# Haptophyte-infecting viruses change the genome condensing proteins of dinoflagellates
Haina Wang [1] ✉, Lingjie Meng [2], Sara Otaegi-Ugartemendia [3], Gabriela Nérida Condezo [3], Romain Blanc-Mathieu [2], Runar Stokke[1,4], Marius Rydningen Langvad [1], David Brandt[5], Jörn Kalinowski[5], Håkon Dahle[1], Carmen San Martín [3], Hiroyuki Ogata [2] & Ruth-Anne Sandaa [1] ✉

Giant viruses are extraordinary members of the virosphere due to their structural complexity and high diversity in gene content. Haptophytes are ecologically important primary producers in the ocean, and all known viruses that infect haptophytes are giant viruses. However, little is known about the specifics of their infection cycles and the responses they trigger in their host cells. Our in-depth electron microscopic, phylogenomic and virion proteomic analyses of two haptophyte-infecting giant viruses, *Haptolina ericina* virus RF02 (HeV RF02) and *Prymnesium kappa* virus RF02 (PkV RF02), unravel their large capacity for host manipulation and arsenals that function during the infection cycle from virus entry to release. The virus infection induces significant morphological changes in the host cell that is manipulated to build a virus proliferation factory. Both viruses' genomes encode a putative nucleoprotein (dinoflagellate/viral nucleoprotein; DVNP), which was also found in the virion proteome of PkV RF02. Phylogenetic analysis suggests that DVNPs are widespread in marine giant metaviromes. Furthermore, the analysis shows that the dinoflagellate homologues were possibly acquired from viruses of the order *Imitervirales*. These findings enhance our understanding of how viruses impact the biology of microalgae, providing insights into evolutionary biology, ecosystem dynamics, and nutrient cycling in the ocean.

Giant viruses belong to a group of viruses distinguished by their exceptionally large size, complex structures and extensive genomic diversity[1]. They constitute the viral phylum *Nucleocytoviricota* and infect a broad range of eukaryotic hosts[2], including multicellular animals, and unicellular protists. Giant viruses are highly abundant and diverse in aquatic environments[3–9], as well as artificial environments of wastewater and terrestrial ecosystems[10]. In the ocean, metagenome analysis has shown that the viruses belonging to the *Algavirales* and *Imitervirales* are the most dominant and widespread families within these phyla[11].

The discovery and investigation of mimivirus infecting amoeba *Acanthamoeba* polyphaga, as well as its relatives[12], has revealed extraordinary features of morphology, genomics and infection cycles. The virions' diameter ranges from ~120 nm[13] to as long as 1 μm[14], exhibiting diverse morphologies, most of which are icosahedral capsids decorated by complex turreted structures and sometimes glycan modifications[1,15]. The general steps in the infection cycle of giant viruses start with membrane fusion and endocytosis to enter the host cell, followed by replication and assembly in the viral factory in the cytoplasm of the host cell or the host nucleus[1]. Mature virions are released after host cell lysis[1]. More intriguingly, these viruses carry a significant array of genes involved in cellular life, encompassing DNA replication and recombination, transcription, translation and posttranslational modifications (PTMs), as well as substantial energy metabolism[1,16–19]. The cellular genes may participate in extensive functions that showcase their extraordinary capability for manipulating their hosts and with potential roles in ecology and global nutrient cycles that are validated by transcriptomic analysis[20–22] and biochemical experiments in vitro[23–25].

We have only a few representatives of viruses in the *Imitervirales* in culture. These are viruses infecting heterotrophic protists such as *Stramenopiles*[18] and *Kinetoplastida*[26], and phototrophic/mixotrophic protists such as Pelagophytes (genus *Aureococcus*[20]), Chlorophytes (genera *Pyramimonas* and *Tetraselmis*[27,28]), and haptophytes (the genera *Haptolina*,

[1]Department of Biological Sciences, University of Bergen, Bergen, Norway. [2]Institute for Chemical Research, Kyoto University, Uji, Japan. [3]Centro Nacional de Biotecnología (CNB-CSIC), Madrid, Spain. [4]Centre for Deep Sea Research, University of Bergen, Bergen, Norway. [5]Bielefeld University, CeBiTec, Bielefeld, Germany. ✉e-mail: haina.wang@uib.no; ruth.sandaa@uib.no

*Prymnesium* and *Phaeocystis*[29]. The viruses infecting haptophytes are an ecologically important group as their hosts are widespread in many oceans, contributing to 30–50% of total chlorophyll biomass[30]. Notably, all the cultured haptophyte viruses so far described are giant viruses. All except one, *Emiliania huxleyi* virus[29], belong to the viral order of *Imitervirales*[31]. Despite the ecological importance of haptophyte viruses, there are few studies focusing on the details of morphological, genomic and proteomic changes during the different steps in their infection.

By the use of genomic, phylogenetic and proteomic analyses, along with transmission electron microscopy (TEM) observation, we have studied in detail the different steps in the viral infection using two giant marine viruses, *Haptolina ericina* virus RF02 (HeV RF02) and *Prymnesium kappa* virus RF02 (PkV RF02), and their hosts as model systems[32]. Our findings show that the viruses harbour a large genome reservoir towards host manipulation that is utilized across the whole infection cycle. Additionally, they hold a pair of the longest recorded inverted terminal repeats (ITRs) that may serve as a hot spot of horizontal gene transfer (HGT). The two viral genomes were found to encode nucleoprotein homologues (i.e. dinoflagellate/viral nucleoproteins, (DVNPs)). Based on phylogenetic analyses across extensive viral metagenomes, combined with structural prediction, we suggest a hypothesis for the evolutionary history of the nucleoproteins of dinoflagellates.

## Results

To address the phylogenetic relationship of HeV RF02 and PkV RF02 with other nucleocytoplasmic large DNA viruses (NCLDVs), we constructed the phylogenetic tree of the DNA polymerase B of *Nucleocytoviricota*. The phylogenetic analysis showed that HeV RF02 and PkV RF02 were assigned to the viral family *Mesomimiviridae* of the order *Imitervirales*[29,32] (Supplementary Fig. 1).

### Morphological changes in host cells across infection cycles

To explore the morphological changes of the host cells due to the viral infection, we compared the virus-free and virus-infected cells by TEM. The uninfected host cells kept intact cellular structures and organelles, including the nucleus, chloroplasts, mitochondria and membrane systems (Fig. 1a, e).

However, upon infection, noticeable changes occurred. During the latent periods, the nucleus and cytoplasm underwent a reorganization to give way to the formation of a viral factory for active virus amplification (at 13 hpi in Fig. 1b and 10 hpi in Fig. 1f). Inside this viral factory, empty capsids of nascent virions were assembled. Subsequently, these capsids underwent maturation as they became filled with viral genetic material (at 24 hpi in Fig. 1c and 13 hpi in Fig. 1g). Finally, the mature virions were released following the cell lysis (Fig. 1d, h). The morphological changes in infected cells clearly illustrate the cellular reorganization from cell growth to virus proliferation.

### Genomic features of HeV RF02 and PkV RF02

HeV RF02 possesses a 582,139-bp genome encoding 545 predicted open reading frames (ORFs), while PkV RF02 harbours a 583,284-bp genome encoding 635 predicted ORFs (Supplementary Data 1). Overall, 36% of ORFs of each genome were functionally annotated (HeV RF02: 198/545, PkV RF02: 226/635). Prediction of transmembrane domains (TMs) and signal peptides (SPs) showed that 23% (HeV RF02: 125/545, PkV RF02: 143/635) of ORFs contained TMs and SPs, of which 77% (96/125) and 73% (105/143) were of unknown function (Fig. 2 and Supplementary Data 1). Functional categories of ORFs are discussed in the section below.

Direct and inverted repeats were distributed across the entire genome in both viruses (Supplementary Data 1 and Fig. 2). Furthermore, imperfect long ITRs were present at the extremities of the viral genomes. The ITRs of the HeV RF02 genome were 96–97 kbp long and those of the PkV RF02 genome were 94 kbp. The ITRs harboured 37% and 36% of the ORFs in each genome, respectively (Fig. 2a, b). The two genomes shared many homologous genes (i.e. 64% for HeV RF02 and 55% for PkV RF02), reflecting their close phylogenetic relationship (Supplementary Fig. 1).

For putative origins of ORFs, virus-originated ORFs constitute the largest set of the two viruses' genes, that is, 61% of HeV RF02 and 56% of PkV RF02, while the ORFans constitute 26% and 33% of the gene set, respectively (Fig. 2f, g). Genes originating from archaea, bacteria and eukaryotes comprise 12% and 10% of the total gene reservoirs, respectively. The rhizomes of the multiple origins are visualised in Fig. 2f, g.

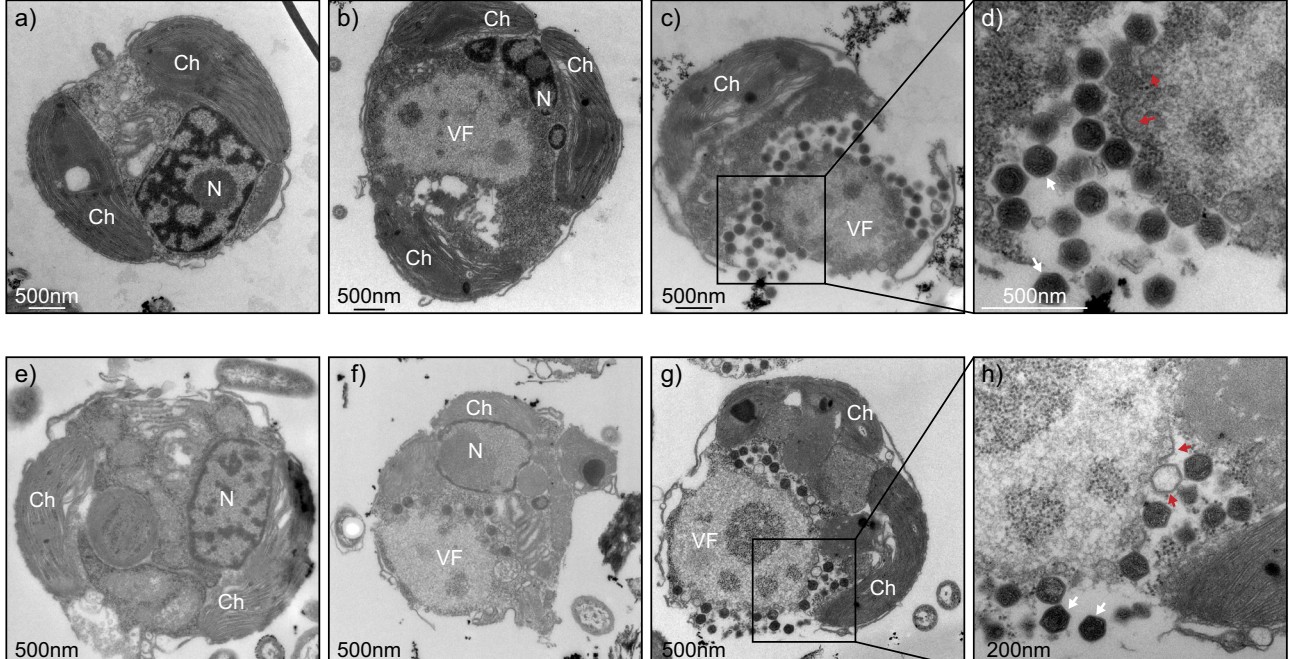

**Fig. 1 | TEM images of infection cycles. a–d** Infection cycle of HeV RF02 infecting *H. ericina* 028. **a** Virus-free host cell. **b)** Infected host cell at 13 h post of the infection (hpi). The viral factory is formed in the cytoplasm. **c** Infected host cell at 24 hpi. **d** Local enlarged image of a3. **e–h** Infection cycle of PkV RF02 infecting *P. kappa* Rcc3423. **e** Virus-free host cell. **f** Infected host cell at 10 hpi. **g** Infected host cell at 13 hpi. **h** Local enlarged image of b3. (**a–d**) show similar cell organization to (**e–h**). Scale bars are indicated on each panel. The red arrows point to the empty capsids and the white arrows point to the mature viruses. Ch chloroplast, N nucleus, VF viral factory.

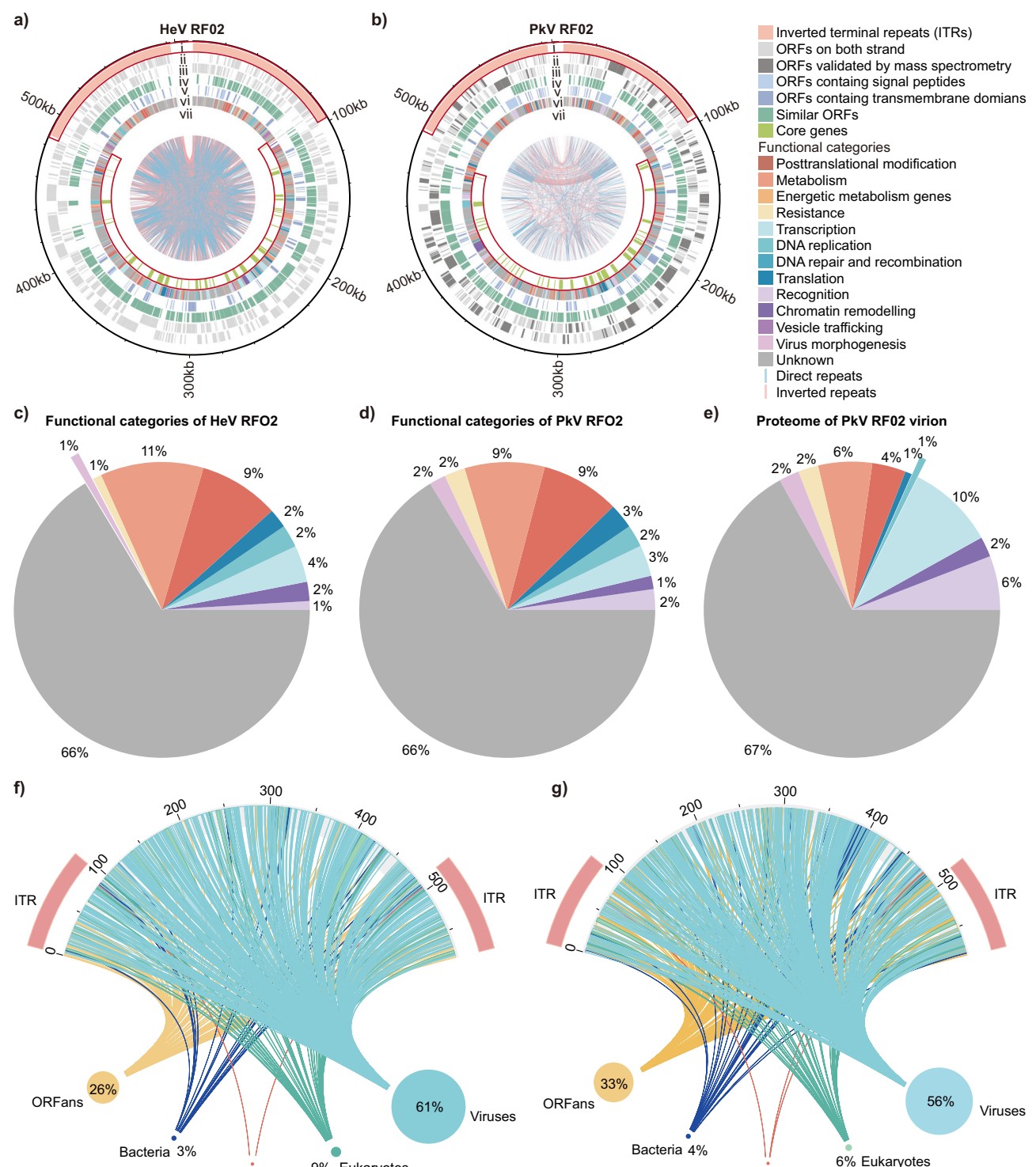

**Fig. 2 | Genome overview of HeV RF02 and PkV RF02.** Genomic features of HeV RF02 (**a**) and PkV RF02 (**b**). Genome sizes are given in kilo base pairs. Nested circles from outermost to innermost correspond to (i) inverted terminal repeats of the two viruses (ITRs); (ii) predicted ORFs on forward strand; and (iii) reverse strand; (iv) homologous ORFs shared by the two viruses; (v) ORFs containing transmembrane and/or signal peptides; (vi) functional classifications of ORFs, with the different functional ORFs indicated by different colours in the legends; (vii) the core genes shared by the flagellated-protist-infecting viruses. The positions of ITRs and core genes are highlighted by red frames to display their relative localisation in the genome maps. The lines in the inner areas display the direct repeats (blue lines) and inverted repeats (red lines). **c** Functional distribution of HeV RF02 ORFs. **d** Functional distribution of PkV RF02 ORFs. **e** Functional distribution of protein contents of PkV RF02 virions. **f** Putative origins of HeV RF02 ORFs. **g** Putative origins of PkV RF02 ORFs.

## Functional categories of ORFs

ORFs are functionally annotated by BLASTp (*E*-value < 1e-5) and hmmsearch (*E*-value < 1e-5), as described in detail in 'Methods'. The biological processes that the gene may be involved in are inferred from the references affiliated with the conserved domains in InterPro[33], as well as the information available in the EcoCyc Database[34]. The functionally annotated ORFs were classified into different steps of the infection cycles, namely virus entry, chromatin remodelling, DNA replication, DNA repair and

**Table 1 | Functional categories of HeV RF02 and PkV RF02 ORFs**

| Functional categories | No. of ORFs | | Representative ORFs and/or pathways |
|---|---|---|---|
| | HeV RF02 | PkV RF02 | |
| Virus entry | 5 | 14 | Concanavalin A-like lectin/glucanases, Dpy-19-like protein, Outer membrane autotransporter, FG-GAP repeat protein |
| Chromatin remodelling | 11 | 9 | DVNP domain-containing protein, Proteins with domains of HMG-box, Cupin-like domain, SWIB, pentapeptide repeat, Sulfolipid-1-addressing protein |
| Transcription | 21 | 21 | DNA-directed RNA polymerase II subunit RPB1, 2, 5, 6, 10, DNA-directed RNA polymerase subunit L376, DNA-directed RNA polymerase subunit D, M, early transcription factor 70 kDa subunit, transcription elongation factor TFIIS_C, late transcription factor VLTF3, VV A18-like intein-containing helicase, TATA-box binding protein, transcription initiation factor IIB, polyA polymerase catalytic subunit |
| DNA replication | 13 | 15 | D5 helicase, topoisomerases, replication factor C (RFC) and putative proliferating cell nuclear antigen (PCNA) |
| DNA repair and recombination | 9 | 12 | Uracil-DNA glycosylase, Exodeoxyribonuclease III, DNA ligase; DNA mismatch repair protein MutS 7, 8; UV damage endonuclease UvdE; ERCC4-type DNA repair nuclease, lambda-type exonuclease, Holliday junction resolvase |
| Translation | 11 | 17 | mRNA capping protein, elF4E, aspartyl-tRNA synthetase, methyltransferase, formyl transferase, tRNA-Ile-lysidine synthetase, Ribonuclease T, XRN family 5'-3' exonuclease, Nudix hydrolases, protein nanos 1 |
| Posttranslational modification | 46 | 53 | Acetyltransferase, methyltransferase, poly [ADP-ribose] polymerase, prenyltransferase, protein kinase, ubiquitin-protein ligases, chaperones, proteases |
| Metabolism[a] | 59 | 54 | Genes involved in the metabolism of nucleotides, amino acids, glycan modification, respiration chain and light harvesting |
| Resistance | 5 | 14 | BAX inhibitor, IAP_GVCPM apoptosis inhibitor IAP, Toll-like receptor, calcineurin-like phosphoesterase, Y domain of phosphatidylinositol-specific phospholipase C |
| Vesicle trafficking | 13 | 6 | Hemagglutinin domain-containing protein, papatin-like phospholipase, t-SNARE family protein, BspA type leucine rich repeat protein, lipid-sensing domain proteins |
| Virus morphogenesis | 5 | 11 | VV A32-like packaging ATPase, major capsid protein, major capsid protein 2, HNH endonuclease |

[a]The category of metabolism is the largest functional groups with genes involved in a broad range of metabolic pathways. Here, we list the suggested pathways for the different ORFs.

recombination, transcription, translation, PTM, vesicle trafficking and virus morphogenesis (Table 1 and Fig. 2a–d). Besides these, genes involved in metabolism and resistance were also identified. The putative roles of the annotated ORFs in the infection cycle are depicted in Fig. 3.

Five and 14 ORFs were classified in the 'Virus Entry' categories of HeV RF02 and PkV RF02, respectively. Most of their protein products are membrane-associated proteins with predicted ligand binding motifs. ORFs with DNA binding domains are classified in the category of 'Chromatin remodelling', namely the ones encoding DVNP domain-containing proteins and those with domains of HMG-box, Cupin-like domain, SWIB, pentapeptide repeat and Sulfolipid-1-addressing protein. The two viruses encoded hypothetical proteins for DNA replication, especially initiation complex, including putative D5 helicase, topoisomerases, replication factor C and putative proliferating cell nuclear antigen. Both viral genomes were found to encode homologous enzymes for DNA base excision repair (BER) and DNA mismatch repair (MMR) pathway (Fig. 3). BER is initiated by uracil-DNA glycosylase (UDG), followed by Exodeoxyribonuclease III (ExoIII, a kind of apurinic/apyrimidinic-endonuclease to generate 3'OH), DNA polymerase to repair the position, and DNA ligase to seal the nick[35]. PkV RF02 encodes a complete set of genes potentially required for the BER pathway, while HeV RF02 encode part of the pathway with the absence of the homologue of UDG. Both viruses encoded homologues of DNA mismatch repair proteins 7 and 8. PkVRF02 encoded a putative UV damage endonuclease (UvdE), with a potential role in repair of UV-irradiated and oxidative damage DNA[36]. Three protein homologues, ERCC4-type DNA repair nuclease, lambda-type exonuclease and Holliday junction resolvase involved in DNA recombination were encoded in both viral genomes. Both viruses harbour a set of putative proteins for a nearly complete transcription apparatus, including essential components of the RNA polymerase complex and various transcription factors. Regarding translation, both viruses encoded putative mRNA capping protein, which is considered to methylate mRNA to generate a 5'-mRNA cap. The viruses also encoded putative elF4E, which functions as mRNA-cap-binding protein. HeV RF02 and PkV RF02 encode 12 and 25 tRNAs, respectively (Supplementary Data 1), while only HeV RF02 encodes one homologue of aspartyl-tRNA synthetase. The two viruses encode a series of translation modification enzymes, including methyltransferase, formyl transferase and tRNA-Ile-lysidine synthetase. Ribonuclease T and XRN family 5'-3' exonuclease, that are considered to facilitate nucleotide recycling, were also encoded. Notably, besides the proteins aiding translation, we found a putative translational repressor protein nano 1 and Nudix hydrolases with a potential function as mRNA-de-capping proteins in both viral genomes.

Additionally, 8% of the ORFs are related to PTMs in HeV RF02 and PkV RF02 each, which constitute the second largest functional group of the two viruses. PTMs participate in extensive biological processes like acetylation, methylation, phosphorylation, prenylation and poly (ADP-ribosyl)ation and ubiquitination (Supplementary Table 1). Moreover, each virus encodes 10 putative chaperones assigned to different families, facilitating protein maturation and complex assembly[37,38]. There are 10 and 11 putative proteases encoded by HeV RF02 and PkV RF02, respectively, which are inferred to function in protein degradation and disaggregation, ubiquitin recycling and virion maturation[39–43]. PkV RF02 may encode a complete ubiquitination system, including one putative E1 ligase (UBA domain-containing protein)[44], and E2 ligase (ubiquitin-conjugating enzyme E2), as well as 13 putative E3 ligases, five of which hold TMs, constituting a hypothetical cascade regulation system with a broad range of substrates and various localisation. HeV RF02 only lacks the putative E1 ligase of the above-mentioned genes in the ubiquitination system.

The largest fraction of annotated ORFs is likely to be involved in metabolism and accounts for 11% (59/545) and 9% (54/635) of the total ORFs in the two viral genomes. Metabolic genes were grouped by their underlying functions as metabolism of nucleotides, amino acids, glycan modification, respiration chain and light harvesting. For nucleic acid biosynthesis, both viruses encode genes possibly involved in both de novo and salvage pathways that are depicted in Fig. 3. The putative dihydrofolate reductase and GMP reductase are encoded for the de novo pathway, and both are probably involved in the synthesis of key intermediate inosine monophosphate. For salvage pathway of conversion from

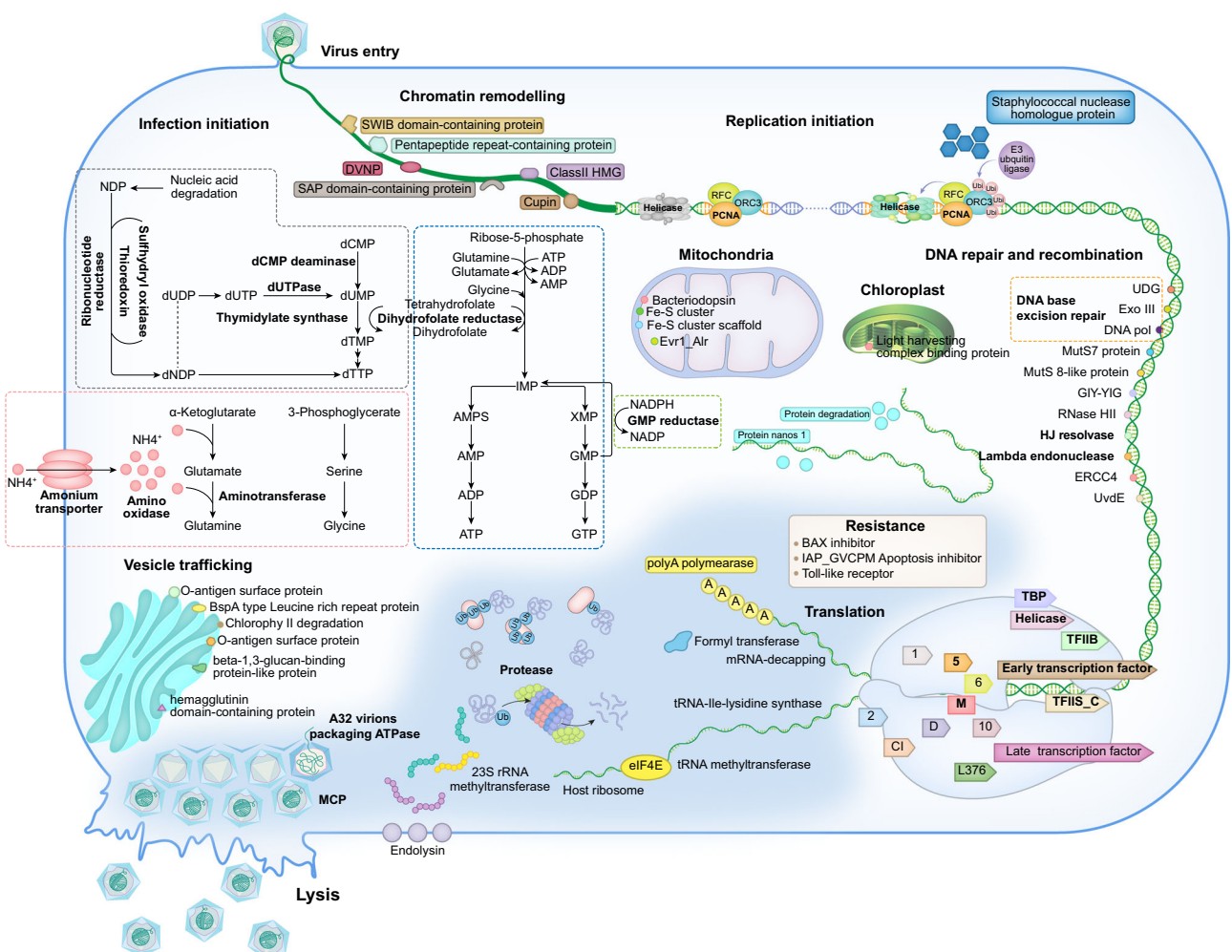

**Fig. 3 | Putative infection cycle predicted from the genome analyses.** The genes involved in virus entry, chromatin remodelling, DNA replication, repair and recombination, transcription, translation, posttranslational modification, vesicle trafficking, metabolism, virus morphogenesis, metabolism and resistance are shown in different colours and the core genes are highlighted in bold font.

NDP to dNDP is hypothetically constructed by the homologues of ribonucleotide reductase, sulfhydryl oxidase and thioredoxin. However, the dTTP must be synthesised from dUDP/dUTP or dCMP. For dUDP/dCMP synthesis, there are five putative enzymes encoded, i.e. dUTPase, dCMP deaminase, thymidylate synthase, deoxynucleoside kinase and dihydrofolate reductase, and the last one might also participate in the de novo biosynthesis, which possibly functions as the linkage of the two pathways. Both viruses also encode a series of homologues of essential enzymes for amino acid synthesis, i.e. putative asparagine synthetase, proline dehydrogenase, arginase, quaternary ammonium transporter and amino-oxidase family protein, gamma-glutamyl hydrolase and aminotransferase. The latter two are only encoded in PkV RF02 of the two viruses. In addition to the amino acid synthesis, the homologue of quaternary ammonium transporter might potentially be involved in nitrogen uptake of host cells in the ocean[23,24]. Rather than encoding enzymes might directly participate in the central carbon metabolism, HeV RF02 and PkV RF02 encode proteins associated with the biosynthesis of coenzymes (2-polyprenylphenol 6-hydroxylase, NAD-binding domain-containing protein), plastoquinol (methyltransferase), Fe-S cluster (Fe-S cluster assembly scaffold protein) and proteins might be involved in cellular redox balance (ERV/ALR sulfhydryl oxidase, disulfide isomerase and putative rhodanese), all of which are critical components in the respiration chain in the mitochondria. There are five genes encoding proteins with a potential function in light harvesting, found in both viral genomes, some of which exist in multiple copies. Two predicted proteins, phytanoyl-CoA

dioxygenase and chlorophyllase, are considered to be involved in chlorophyll degradation. Bacteriorhodopsin-like proteins are encoded by both HeV RF02 and PkV RF02, and they are predicted to function as a proton pump for either energy conversion or ion channel. Putative genes involved in glycosylation were identified from both viral genomes. The genes encoding glutamine-fructose-6-phosphate transaminase, glucosamine 6-phosphate N-acetyltransferase and N-acetylglucosamine-1-phosphate uridyltransferase constitute a nearly complete biosynthesis pathway of uridine diphosphate N-acetylglucosamine (UDP-GlcNAc), with the absence of phosphoglucomutase 3. In addition to synthesis of UDP-GlcNAc, multiple enzymes involved in nucleotide-sugars metabolism are encoded in HeV RF02, like GDP-mannose and dTDP-D-glucose. There are 7 and 5 glycosyltransferases (GTs) encoded in the genomes of HeV RF02 and PkV RF02, respectively, for glycosylation of a wide range of substrates.

The two viruses may also encode membrane surface proteins (hemagglutinin domain-containing protein) and lipid modification and hydrolysis (papatin-like phospholipase) potentially involved in vesicle trafficking. Additionally, some unique protein homologues, like t-SNARE family proteins, BspA type leucine rich repeat proteins and lipid-sensing domain proteins, are also included in the functional group of vesicle trafficking.

For viral resistance against the host immune system, we identified three distinct sets of genes. These code for DNA methyltransferase responsible for protecting viral DNA from host degradation, inhibitor of apoptosis and

proteins of general functions in signal transduction. These inhibitors are transmembrane BAX inhibitor, IAP_GVCPM apoptosis inhibitor IAP and Toll-like receptor.

Four genes are classified in the functional group of 'Virus morphogenesis' in both viral genomes. They encode two types of major capsid proteins, A32-like packaging ATPase and HNH endonuclease. The HNH endonuclease is a key component of the genome packing apparatus in tailed phages[45]. Six HNH copies were found in the genome of PkV RF02, four of which are located in the ITRs.

## Proteome analysis of PkV RF02 virions

The protein content of purified PkV RF02 particles was determined by mass spectrometry, which revealed 136 of 635 ORF products present in the virions, distributed across the entire viral genome (Supplementary Data 2, gene positions are marked in Fig. 2b). The functions of these proteins were distributed extensively in virus entry (6%), chromatin remodelling (2%), DNA replication (1%), transcription (9%), translation (1%), PTMs (4%), metabolism (6%), resistance (2%) and virus morphogenesis (2%) (Table 2 and Fig. 2e). Ninety-one ORFs are of unknown function (Supplementary Data 2). Intriguingly, proteins coding for a nearly complete transcriptional apparatus were packaged in the virions, in addition to a DVNP domain-containing protein (pI value = 10.7).

## Core genes

Only 8% (44/545) and 7% (43/635) of the ORFs in HeV RF02 and PkV RF02, respectively, are core genes shared by the group of flagellated-protist-infecting viruses, and more than half of them code for proteins that are packaged in the PkV RF02 virions. Most of the core genes are located in the central region of the genomes, distributed in the functional groups of DNA replication, recombination and repair, transcription, translation and nucleotide metabolism, closely related to the process of genetic information storage, maintenance and expression (Supplementary Table 2). Six of the core genes encode hypothetical proteins and three of them are packaged in the PkV RF02 virions.

## Phylogenetic analysis of DVNPs

A DVNP is a viral-derived protein acquired by dinoflagellate as the nucleoprotein replacing histones and represents a remarkable exception of eukaryotic chromatin proteins[46]. The discovery of DVNP homologues in HeV RF02 and PkV RF02 prompted further investigation of the distribution of these homologues in related giant viruses. To explore the distribution of DVNPs in the group of marine giant viruses, we generated an HMM file based on known DVNPs to search for their homologues against the Global Ocean Eukaryotic Viral (GOEV) database[47]. PSI-BLAST was also executed to detect the distant DVNP homologues in the database. This database contains 1817 genomes/viral metagenome-assembled genomes (vMAGs) that represents a comprehensive repository for giant viruses in marine environments[47]. In total, 638 sequences were retrieved by the two searches, and 528 (29%) viral genomes/vMAGs were found to encode DVNP homologues, primarily distributed in two viral orders, *Imitervirales* (N = 419; 38%) and *Algavirales* (N = 98; 43%) (Fig. 4a). While a majority of these viral genomes harbour a single copy of a DVNP gene, a specific lineage within *Imitervirales* encodes multiple copies (Fig. 4a). Within the order of *Algavirales*, prasinoviruses and phaeoviruses encode DVNP homologues, whereas chloroviruses do not. Viruses with large genomes, such as mimiviruses and pandoraviruses infecting amoeba, lack DVNP homologues. Very few detections of DVNP homologues were identified in viruses from genomes belonging to *Asfuvirales*, *Pimascovirales* and *Pandoravirales*.

To investigate the origin and evolution of DVNPs, we expanded the PSI-BLAST search (10 iterations, E-value < 1e-3) to the NCBI non-redundant protein sequences database, to identify more DVNP homologues. This search resulted in 264 hits, most of which were predominantly from genomes belonging to dinoflagellates. The resultant sequences, combined with DVNP homologues identified in the GOEV database, were used

to construct the phylogenetic tree (Fig. 4a). Additionally, the sequences were filtered, and the sequence alignment was refined for phylogenetic tree construction, as described in Methods. The phylogenetic tree of DVNP sequences is predominantly separated into two major subclades, i.e. *Imitervirales* and *Algavirales* (Fig. 4b). Notably, the cellular sequences retrieved from the non-redundant database constituted a distinct monophyletic clade, yet nested within the *Imitervirales* clade, with the DVNPs of phaeoviruses (*Ectocarpus siliculosus* virus (EsV) and *Feldmannia* sp. virus (FsV)) in the order of *Algavirales*. Our results showed that DVNP homologues are prevalent in marine giant viruses, particularly within the orders of *Imitervirales* and *Algavirales*. Our phylogenetic analysis revealed that the cellular DVNPs are only found in dinoflagellates and points to their potential origin from viruses of *Imitervirales*.

To investigate whether the cellular DVNPs are able to enter the nucleus to replace the histones we examined whether these DVNPs contain the nucleus localisation signal (NLS) that mediates the transport of proteins synthesised in the cytoplasm into the nucleus. NLStradamus based on the hidden Markov model (HMM)[48] was applied to predict the NLS of the DVNPs (cutoff value of 0.9). We found that 672 out of 718 DVNPs contained NLS. The multiple sequence alignment (Supplementary Data 3) and the predicted NLSs (Supplementary Data 4) are displayed following the rectangular phylogenetic tree of DVNPs (Supplementary Fig. 2). Curiously, most viral homologues of DVNPs harbour NLS at C-terminus, while DVNPs of dinoflagellate harbour NLS close to the N-terminus. Both termini contain highly variable regions in the DVNP sequences. Hence, the NLSs are mostly located in the highly variable regions. The crystal structure of any DVNP remains unexplored, due to the difficulty in protein expression of DVNPs, of which protein products form insoluble inclusion bodies in the expression cells[49]. Given the challenges associated with DVNP expression, the structural prediction is essentially significant as it provides additional structural information on DVNPs. To gain insights into the NLS localisation in the 3-D structures, we predicted the 3-D structures of the two DVNP protein sequences (ORF414 of PkV RF02 and DVNP.5) and highlighted their predicted NLSs (Fig. 4c). As a result, the localisation of the predicted NLS differed between the viral and dinoflagellate DVNPs at the levels of both protein sequences and 3-D structures. This implies that the predicted NLSs might be acquired from different sources.

## Discussion

Histones are fundamental elements for eukaryotic cells to package their genomic DNA molecules into chromatin. Dinoflagellates are a remarkable exception, as some of them utilise DVNP as the nuclear protein, instead of histones[49,50]. Dinoflagellate DVNPs have been suggested to originate in algal viruses (i.e. phaeoviruses of the order *Algavirales*)[49]. However, the function of the viral homologues is still unknown and the distribution of homologues in viruses has not been well-characterised. Here, the existence of DVNP in the PkV RF02 virions was validated by mass spectrometry (Fig. 2 and Table 2). Given the positive charge of the protein (pI value 10.7), it is likely that the DVNP is bound to the viral genome inside the capsid. This indicates that the DVNP probably functions as the DNA-binding protein of the virus. Furthermore, we discovered that DVNP homologues are widespread in marine giant viruses, including viruses of *Imitervirales* and *Algavirales*, rather than being restricted to phaeoviruses. Contrary to the previously suggested phaeovirus (*Algavirales*) origin of DVNP, our phylogenetic analyses suggest that DVNP was derived from viruses within the *Imitervirales* order. We also found that most DVNPs in this study contained NLS, which are usually located in the highly variable regions in these sequences. Sequence and structure prediction comparisons show that the NLSs of the cellular DVNPs are not conserved in viral homologues, indicating that the NLSs of the cellular DVNPs do not originate in viral homologues. Hence the dinoflagellates may acquire DVNPs from viruses of *Imitervirales*. Furthermore, their NLSs directing the localisation in the nucleus might have been generated from sources other than viruses belonging to *Imitervirales*, which could have contributed to the replacement of histones.

**Table 2 | PkV RF02 virus proteins identified by mass spectrometry in purified virions[a]**

| ORF number | Description | MW [kDa] | pI | Replicate 1 | | Replicate 2 | |
|---|---|---|---|---|---|---|---|
| | | | | Coverage [%] | Abundances [%] | Coverage [%] | Abundances [%] |
| **Virus entry** | | | | | | | |
| ORF267 | Concanavalin A-like lectin/glucanases | 37.4 | 6.7 | 72 | 1.31 | 72 | 1.54 |
| ORF268 | Concanavalin A-like lectin/glucanases | 33.3 | 9.13 | 52 | 0.89 | 47 | 0.97 |
| ORF195 | Extracellular link domain-containing protein | 30.7 | 7.18 | 29 | 0.34 | 32 | 0.25 |
| ORF496 | Putative lipoprotein outer membrane proteins with heavy-metal binding domain | 14.2 | 10.13 | 42 | 0.26 | 22 | 0.07 |
| ORF413 | Outer membrane autotransporter | 23 | 5.31 | 25 | 0.05 | 16 | 0.02 |
| ORF269 | Concanavalin A-like lectin/glucanases | 55.6 | 8.62 | 20 | 0.04 | 19 | 0.09 |
| ORF473 | VOG03348; REFSEQ FG-GAP repeat protein | 139.6 | 4.39 | 5 | 0.03 | 2 | 0.01 |
| **Chromatin remodelling** | | | | | | | |
| ORF435 | Pentapeptide repeat-containing protein | 170.9 | 8.87 | 45 | 2.52 | 40 | 3.08 |
| ORF414 | DVNP domain-containing protein | 11.9 | 10.7 | 27 | 0.72 | 10 | 0.24 |
| ORF434 | Pentapeptide repeat-containing protein | 156.7 | 8.24 | 26 | 0.35 | 24 | 0.36 |
| **Replication** | | | | | | | |
| ORF84 | NAD-dependent DNA ligase | 129.7 | 9 | 29 | 0.29 | 30 | 0.25 |
| **Transcription** | | | | | | | |
| ORF204 | Early transcription factor 70 kDa subunit | 119.7 | 9.31 | 54 | 1.23 | 50 | 2.18 |
| ORF138 | Early transcription factor 70 kDa subunit | 98.4 | 8.48 | 59 | 0.69 | 62 | 1.10 |
| ORF238 | DNA-directed RNA polymerase II subunit RPB2 | 168.6 | 5.33 | 32 | 0.58 | 38 | 0.91 |
| ORF245 | DNA-directed RNA polymerase II subunit RPB1 | 173.1 | 6.7 | 29 | 0.50 | 35 | 0.95 |
| ORF273 | Early transcription factor 70 kDa subunit | 158.7 | 6.04 | 42 | 0.47 | 37 | 0.74 |
| ORF361 | DNA-directed RNA polymerase II subunit D | 43.2 | 6.46 | 37 | 0.24 | 39 | 0.27 |
| ORF452 | polyA polymerase catalytic subunit | 53.5 | 9.44 | 43 | 0.18 | 29 | 0.22 |
| ORF237 | DNA-directed RNA polymerase II subunit RPB5 | 24.9 | 6.6 | 31 | 0.16 | 32 | 0.15 |
| ORF274 | DNA-directed RNA polymerase subunit L376 | 18.4 | 5.24 | 38 | 0.12 | 33 | 0.10 |
| ORF226 | DNA-directed RNA polymerase subunit M like protein | 15 | 7.15 | 24 | 0.08 | 37 | 0.05 |
| ORF225 | DNA-directed RNA polymerase II subunit RPB6 | 29.9 | 4.35 | 36 | 0.07 | 21 | 0.04 |
| ORF241 | DNA-directed RNA polymerase, subunit N/RPB10 | 9.1 | 8.7 | 39 | 0.06 | 43 | 0.04 |
| **Translation** | | | | | | | |
| ORF62 | Putative Nudix hydrolase | 19.9 | 9.07 | 21 | 0.02 | 7 | 0.01 |
| ORF277 | mRNA capping enzyme | 141.4 | 8.51 | 30 | 0.38 | 39 | 0.72 |
| **Posttranslational modification** | | | | | | | |
| ORF157 | EF-hand-containing protein | 18.9 | 9.67 | 33 | 0.19 | 22 | 0.11 |
| ORF213 | Metal-dependent hydrolase | 22.9 | 8.78 | 48 | 0.15 | 37 | 0.22 |
| ORF134 | SUMO-1-specific cysteine protease | 46.2 | 9.86 | 20 | 0.06 | 26 | 0.18 |
| ORF93 | Protein kinase | 59.2 | 8.66 | 14 | 0.06 | 8 | 0.07 |
| ORF442 | Phosphoesterase PA-phosphatase related protein | 22.4 | 9.47 | 34 | 0.03 | 25 | 0.03 |
| **Metabolism** | | | | | | | |
| ORF327 | Gamma-glutamyl hydrolase | 35.8 | 9.98 | 37 | 0.59 | 42 | 0.59 |
| ORF155 | Methyltransferase | 39 | 8.13 | 54 | 0.31 | 45 | 0.34 |
| ORF379 | Disulfide isomerase (thioredoxin domain-containing) | 18.8 | 5.4 | 36 | 0.13 | 33 | 0.25 |
| ORF352 | O-antigen ligase like membrane protein | 13.8 | 9 | 24 | 0.13 | 43 | 0.08 |
| ORF439 | Putative rhodanese | 33.5 | 9.77 | 31 | 0.13 | 39 | 0.12 |
| ORF144 | Bacteriorhodopsin-like protein | 26.2 | 8.46 | 11 | 0.08 | 11 | 0.12 |
| ORF440 | Amino-oxidase family protein | 52.3 | 9.45 | 21 | 0.06 | 26 | 0.13 |
| ORF378 | ERV/ALR sulfhydryl oxidase | 23.4 | 9.96 | 19 | 0.06 | 15 | 0.18 |
| **Resistance** | | | | | | | |
| ORF451 | Phosphatidylinositol-specific phospholipase C, Y domain | 46.3 | 8.1 | 46 | 0.31 | 40 | 0.27 |
| ORF75 | Putative Calcineurin-like phosphoesterase | 52.4 | 9.01 | 35 | 0.18 | 19 | 0.14 |

**Table 2 (continued) | PkV RF02 virus proteins identified by mass spectrometry in purified virions[a]**

| ORF number | Description | MW [kDa] | pI | Replicate 1 | | Replicate 2 | |
|---|---|---|---|---|---|---|---|
| | | | | Coverage [%] | Abundances [%] | Coverage [%] | Abundances [%] |
| ORF312 | Putative ion channel domain-containing protein | 18.4 | 6.57 | 31 | 0.02 | 29 | 0.04 |
| **Virus morphogenesis** | | | | | | | |
| ORF313 | Major capsid protein MCP | 58.1 | 5.1 | 56 | 15.41 | 62 | 35.74 |
| ORF450 | Major capsid protein MCP2 | 66.2 | 5.95 | 37 | 0.18 | 45 | 0.23 |
| ORF270 | HNH endonuclease | 22.8 | 9.48 | 40 | 0.01 | 19 | 0.02 |

[a]The detected viral proteins of PkV RF02 virions with clear annotation are summarised in Table 2. The full results table for mass spectrometry is attached as Supplementary Data 2.

Due to their large genome size, compacting genomes within the capsids of giant viruses poses significant challenges. Some giant viruses utilise structural proteins to condense their expansive genomes[51]. In addition to DVNPs, various giant viruses compact their genomes using histone-like repeats. These viral histones are found in marseille-viruses, iridoviruses, medusaviruses and some deep-branching viral superclades[52]. It has been proposed that these viral histones represent evolutionary intermediates between archaeal and eukaryotic nucleosomes[52–55]. Although both viral DVNPs and histones are involved in genome condensation, they exhibit no similarity at the amino acid sequence level. Beyond condensation via DVNPs and histones, Mimivirus employs a unique mechanism to compact its 1.2-Mb genome. This involves a nucleoprotein fibre, which is subsequently encased within a 30-nm diameter helical protein shell[56]. The essential proteins involved in Mimivirus genome condensation remain unknown. These diverse genome-condensing mechanisms observed in giant viruses suggest that their condensation strategies may have evolved independently.

The two viruses described here harbour the longest ITRs per genome size in the flagellated-protist-infecting virus group of the order *Imitervirales*[26]. The employment of the nanopore sequencing method enhanced the reliability of the long ITRs. Although repeat sequences are enriched in the ITRs, functional proteins are evenly encoded in this region, demonstrated by proteomic analysis of PkV RF02 virions. Most core genes are located in the central region of the genomes, indicating that genes located in ITRs are less conserved.

Many giant viruses harbour ITRs ranging from hundreds of base pairs to 23 kb that function in various ways[26,57–59]. The 900-bp ITRs of Mimivirus may allow a circular topology of the annealing[58]. In addition to the functions of the short ITRs in the viral genome topology and replication, long ITRs of giant viruses appear to be involved in the gene acquisition leading to genome expansion. All the long ITRs mentioned, including those from HeV RF02, PkV RF02, poxviruses[60], *Bodo saltans* virus (BsV)[26] and Megavirus Baoshan[57], include not only repeated elements, but also hypothetical genes with putative essential functions. Some of these genes are expressed and can be detected in both transcriptomic and proteomic analyses[57]. In the ITRs of BsV, the enrichment of ankyrin-repeat domains suggests a recent sequence duplication expanding the viral genome[26]. Similarly, in poxviruses, such as Mpox viruses causing global outbreaks in recent years, extensive ITR expansion has been observed, involving gene duplication and loss[61]. These genes may function in immune modulation, virulence and host adaptation[61]. In the model poxvirus vaccinia, studies in human cells have shown that the recombination-mediated gene expansions can facilitate acquisition of adaptive genetic elements of viruses[62]. These findings indicate that gene acquisition in ITRs may play roles in the poxivirus adaptation of the host antiviral defenses[61,62]. Additionally, extensive lateral acquisition of genes from hosts and bacterial symbionts has been detected by phylogenetic analysis in many NCLDV lineages[63]. The larger the genome, the higher the number of laterally acquired genes[63]. Interestingly, although it remains unknown whether these viral genomes contain ITRs, all of these laterally acquired genes show a strong tendency to be positioned at the extremities of the viral genomes[63]. This pattern aligns with the gene distribution in viral genomes of HeV RF02 and PkV RF02, where genes located in ITRs are less conserved compared to those in the central region of the genome. Based on

the presence of enrichment of repeat sequences, and the functional genes validated by the proteomic analyses, combined with the above analyses, it is tempting to suggest that ITRs of HeV RF02 and PkV RF02 might be a potential region for HGT from their hosts; and that gene acquisition by the long ITR may be the way of genetic innovation or genome inflation in giant viruses.

Although only one-third of the genes in the two viral genomes were annotated, these genes exhibited a large capacity for host manipulation and efficient infection strategies across the whole infection cycle. Both viruses encode putative proteins of the initiation complex of DNA replication and most of them are core gene products of the flagellated-protist-infecting virus group. It is challenging to replicate giant virus DNA in the cytoplasm, due to the difficulties in precise replication of long DNA molecules and mutations resulting from spontaneous chemical reactions[35]. The viruses encoded two hypothetical DNA repair pathways that are DNA BER and DNA MMR (Fig. 3). This might be the strategy to ensure the fidelity of DNA replication in the host cytoplasm. UvdE is encoded by PkV RF02, which is involved in the repair of UV-irradiated and oxidative damage DNA[36], adaptive for its algal host living on the marine surface. Moreover, the two viruses encode a nearly complete transcriptional machinery, which was detected by mass spectrometry in the PkV RF02 virions. The transcription apparatus packaged in the virions allows the quick start of transcription, aiding efficient viral hijacking. This remarkable strategy is also adopted by Acanthamoeba-infecting viruses[56] and mammalian dsDNA poxvirus[64], as well as dsRNA reovirus[65]. Instead of the autonomous translational apparatus encoded by other giant viruses[26,66], both HeV RF02 and PkV RF02 only encode the crucial components of translation initiation and RNA modifications. Viral mRNA can be methylated by core-gene-encoded mRNA-capping proteins, followed by the binding of eukaryotic initiation factor 4E (eIF4E)[67]. The eIF4E binding to 5′-mRNA cap is a rate-limiting determinant of translation, leading to the recruitment of translational complex[68], which can be regulated by phosphorylation[69]. On the contrary, the host mRNA can be de-capped by multiple copies of Nudix hydrolases functioning as mRNA-de-capping proteins[70], and Protein Nanos 1 functions as the translational repressor by binding to the 3′untranslated region (3′UTR) of mRNA[71]. For RNA molecules, various nucleases and methyltransferase, formyl transferase and tRNA-Ile-lysidine synthetase are encoded for mRNA degradation[72], tRNA maturation[73] and end-turnover of deficient tRNAs[74], facilitating nucleotide recycling.

From bacteria, plants and vertebrates, host cells have evolved sophisticated immune systems to defend against viral infections, leading to DNA degradation, nucleotide depletion, ER (endoplasmic reticulum) stress and programmed cell death, such as apoptosis[75,76]. Meanwhile, viruses develop corresponding strategies to inhibit the host's immune responses and promote infection. Here, both HeV RF02 and PkV RF02 encode multiple copies of DNA methyltransferases that may protect the viruses from host degradation[77]. The high consumption of nucleotides is supported by the virus-encoded nucleotide metabolic enzymes to overcome the host defence against nucleotide depletion[75] (Fig. 3). All the genes involved in the conversion from NDP to dNDP are core genes of the group of flagellated-protist-infecting viruses, implying their essential roles in viral infection (Supplementary Table 2).

a)

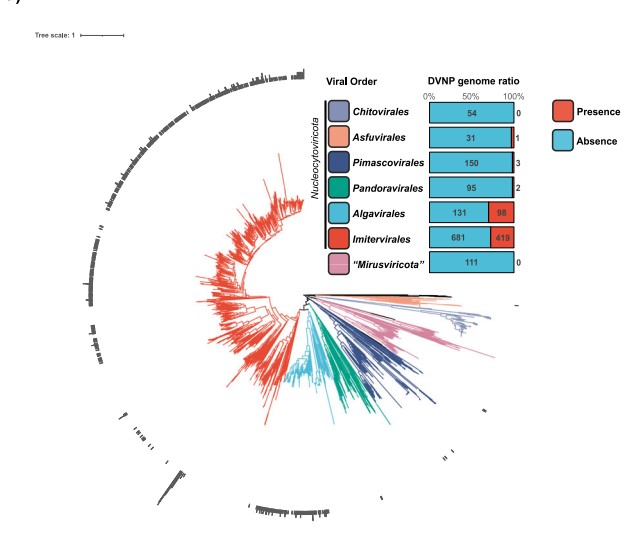

b)

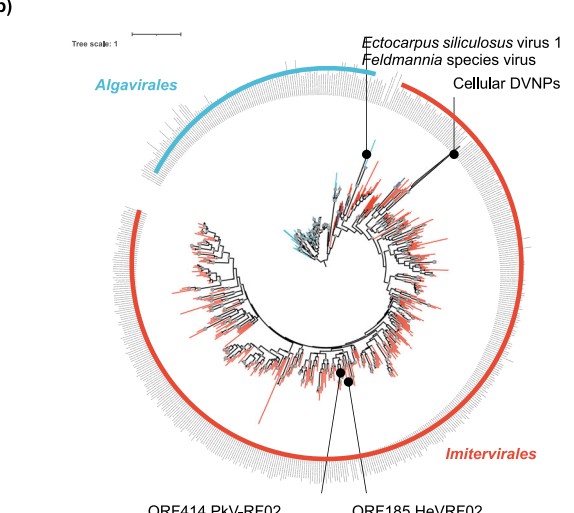

c)

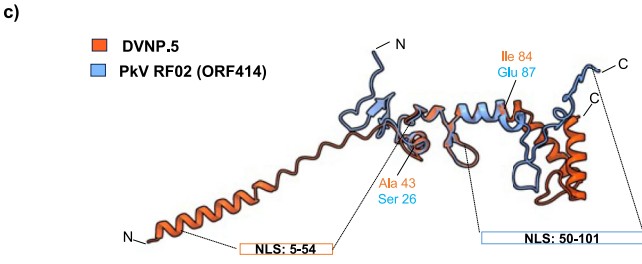

**Fig. 4 | Distribution and phylogenetic tree of DVNPs. a** Distribution of DVNP in marine giant viruses displayed in the phylogenetic tree constructed by protein sequences of DNA polymerase B. The outer circle displays the distribution of DVNP in marine giant viruses. **b** Phylogenetic tree of DVNP homologues. Local support values ≥85 were shown by grey circles. **c** Comparison of Alphafold2.0 predictions of DVNP representatives from dinoflagellates (DVNP.5) and *Imitervirales* (PkV RF02 ORF414). The predicted NLSs are highlighted by dashed lines, showing that the NLSs of PkV RF02 and DVNP.5 locate in different positions of the proteins. The start and end residues of the structurally aligned region are indicated by labelling the corresponding residues (Supplementary Fig. 3).

The formation of a viral factory is the primary morphological trait of infected cells inside which virions are replicated and assembled actively (Fig. 1). The excessive viral protein production causes the accumulation of unfolded and misfolded peptides in the ER that triggers the unfolded protein response (UPR) to restore the ER function[78]. When the regulation capacity of UPR signalling is overwhelmed, the host cell undergoes ER stress, inducing subsequent apoptosis[78,79]. However, viruses cannot benefit from host cell apoptosis. HeV RF02 and PkV RF02 develop two aspects as strategies to overcome ER stress and inhibit apoptosis. On the one hand, the viruses encode chaperones and proteases accompanying ubiquitination systems that participate in peptide folding, disaggregation and degradation to restore the ER's normal function (Supplementary Table 1). On the other hand, BAX inhibitor encoded by PkV RF02 functions via interaction with calmodulin to inhibit the cell death triggered by ER stress[80]. Except for BAX inhibitor, apoptosis inhibitor IAP_GVCPM and Toll-like receptors encoded by PkV RF02 and HeV RF02, respectively, suppress the apoptosis via multiple targets in the complicated immune pathways[81,82].

The viruses in this study regulate haptophytes, an ecologically significant and highly diverse group of marine algae that include both bloom-forming and non-bloom-forming species[29]. The structure of haptophyte communities in marine ecosystems is influenced not only by physico-chemical factors but also by viral infections[29]. Through host cell lysis, viruses affect not only the abundance of haptophytes but also indirectly influence biogeochemical cycles[83] and the efficiency of the biological carbon pump[84].

Our study offers insights into the intricate strategies that viruses employ to manipulate host populations throughout the entire infection cycle, from host cell entry to the release of nascent viral particles. Furthermore, we propose that ITRs might facilitate HGT between giant viruses and their host. This hypothesis is supported by the observation that laterally transferred genes are often located at the extremities of the viral genomes. A notable example of HGT identified in our study is the viral origin of dinoflagellate DVNPs, which likely results from gene exchange between giant viruses and hosts. These findings significantly enhance our understanding of the evolutionary history of these giant viruses and their hosts, as well as the potential molecular mechanisms underlying their interactions.

## Methods

### Transmission electron microscopy (TEM)

The images of virus-free and virus-infected host cells were acquired from thin-section samples. Samples were fixed in 2.5% glutaraldehyde (Merck KGaA D-64217) for at least 24 h at 4 °C. The fixed cells were harvested by centrifugation at 5500 rpm for 20 min (Beckman Coulter Avanti JXN-26, rotor JS-7.5) and resuspended in fresh ½ IMR media[32]. Post-fixation was performed for 1 h (on ice) in 1% osmium tetroxide (EMS # 19134) diluted in 0.1 M sodium cacodylate buffer, followed by twice washing. The samples were then dehydrated using a graded ethanol series (30%, 50%, 70%, 96% and 100%) before transfer to a 1:1 solution of 100% ethanol:propylene oxide for 15 min. They were then transferred to 100% propylene oxide for 15 min with gradual introduction of agar 100 resin (AgarScientific R1031) drop by drop over the next hours. After that, samples were transferred to a small drop of 100% resin and excess propylenoxid was allowed to evaporate for 1 h. Afterwards the samples were transferred to 100% resin and placed in moulds and left at room temperature overnight. The moulds were placed at 60 °C for 48 h to polymerise. Ultra-thin sections of approximately 60 nm were placed on 100-mesh formvar-coated (EMD # 15820) copper grids (EMS #G100H-Cu) and stained with 2% uranyl acetate (EMS # 22400) and lead citrate (VWR #1.07398). Grids were imaged by Hitachi HT7800 TEM at 120 kV or Jeol JEM-1230 TEM at 80 kV.

### Purification of viral particles and DNA isolation

Exponentially growing host cultures, namely *Haptolina ericina* UiO028 and *Prymnesium kappa* UIO034 (2 L)[32], were infected with 20-mL infectious lysates of HeV RF02 and PkV RF02, respectively. An uninfected culture (100 ml) was used as a control. For viral particles purification and DNA isolation we in detail followed the protocol described

in Blanch Mathieu et al.[16]. The host strains used in this study were obtained from Prof. Bente Edvardsen of University of Oslo[32].

## Genome sequencing and assembly

Isolated DNA from PkV RF02 and HeV RF02 was subject to transposase-based Rapid nanopore library preparation (SQK-RBK004) and sequenced on an Oxford Nanopore GridION device and a FLO-MIN106 flow cell, resulting in 47,551 and 71,637 long reads with a total length of 154.48 Mbp and 145.39 Mbp, respectively. Illumina TruSeq PCR-free libraries were prepared as well (insert size, 350 bp). The generated libraries were sequenced on an Illumina MiSeq instrument in paired-end mode (2 × 300 bp) to yield ~1.5 million and 1.4 million reads, respectively. Long reads were assembled to 492,719 bp and 487,285 bp in size using Canu v1.7.1[85]. For both viruses, assemblies contained two contigs, where one contig corresponded to terminally repeated sequences of about 90 kb at both ends of the genomes, and one to the genomic region in between the two terminal repeats. Assemblies were then manually refined by adding the terminal repeat sequences to both genome ends and were polished using long-reads with Nanopolish (https://github.com/jts/nanopolish) and short-reads with Pilon[86] to yield final linear genome sequences of 583,284 bp for PkV RF02 and 582,139 bp for HeV RF02.

## Functional annotation and classification of predicted ORFs

ORFs were predicted by GeneMarkS with the option 'virus'[87] in the assembled genome sequences of HeV RF02 and PkV RF02, respectively. tRNAscan web server (http://lowelab.ucsc.edu/tRNAscan-SE/) was applied to identify tRNAs using the default mode[88,89]. Functional annotation of each ORF is defined manually based on full-length alignment of BLASTp and match of conserved domains, described as follows. Firstly, amino acid sequences of ORFs were searched against public databases, including Virus-Host DB[90], RefSeq[91], UniRef90[92] and COG[93] with an E-value cutoff of 1e-5. The annotations were further confirmed by conserved domains identified by hmmsearch (hmmer.org) against databases of Pfam[94], VOGDB[95] (https://vogdb.org/; VOGDB release 212), as well as InterPro[33] with an E-value cutoff of 1e-5. In ambiguous cases, for example when using hmmsearch to identify conserved domains in protein sequences, we may obtain several different hits that meet the threshold, some of which overlap. In such instances, we select the hits with the lowest E-value in the same region in the protein sequences, to minimise the overlap between identified domains. The functional annotations were also checked manually through an integral online tool, GenomeNet (https://www.genome.jp/; alignment quality, length comparison to canonical genes, and links with KEGG orthology). Giant viruses shared a largely conserved life cycle[1,12,51], including the steps of virus entry, chromatin remodelling, DNA replication, DNA repair and recombination, transcription, translation, PTM, vesicle trafficking and virus morphogenesis. To understand how viruses infect and manipulate their hosts using their genetic reservoir, we classified the annotated genes into functional groups corresponding to the steps of the infection cycle mentioned above. The classification is based on the functional annotation of ORFs, combined with functional information inferred from the references affiliated with the conserved domains in InterPro[33], as well as the information available in the EcoCyc Database[34]. To expand the functional annotation of hypothetical proteins with unknown function, we predicted SPs and TMs using SignalP6[96] and DeepTMHMM[97] using default settings. For some types of putative enzymes that may function in extensive biological processes, we searched for information in specialised databases to infer their functions. The functional classification of GTs and proteases was defined by the databases CAZy[98] and MEROPS[99], respectively. The metabolic pathways were generated based on the information from the EcoCyc database[34].

## Phylogenetic analysis of DNA polymerases and DVNPs

Protein sequences of DNA polymerases were collected from the GOEV database[47]. These sequences were subsequently used to construct the maximum-likelihood phylogenetic tree. −m TEST was used for the model selection and pfam+F+I+G4 was chosen according to BIC (Bayesian Information Criterion). The tree was rooted using the *Algavirales* clade.

We used DVNP sequences from HeV RF02 and PkV RF02, together with previously reported viral DVNPs from EsV and FsV, as well as reported cellular DVNPs (DVNP.5, 10, 12, 13) of dinoflagellate *Hematodinium* sp[49]. These sequences were used to execute multiple sequence alignment by MAFFT—linsi. To obviate the influence of non-conservative regions, columns with more than 60% gaps were excluded using Trimal. This generated a refined DVNP-HMM file, which was then used for hmmsearch in the following steps. To investigate the distribution of DVNP homologues in marine giant viruses, we used the DVNP-HMM file generated in the previous step to perform hmmsearch (E-value < 1e-3) against the GOEV database. This database contains 1817 genomes/metagenomic assembled genomes (MAGs), and represents a comprehensive repository for giant viruses in marine environments[47]. Additionally, we also used the DVNP sequence from dinoflagellate (AFY23224.1) as the seed for PSI-BLAST (10 iterations, E-value < 1e-3) against the GOEV database, due to its high average identity with other DVNP sequences.

To explore the origin and evolution of DVNPs, the search for DVNP homologues was expanded to a non-redundant sequence database of GenBank[100] with the same query parameters and threshold as described above. In total, 910 homologue sequences of DVNP were obtained by hmmsearch and PSI-BLAST against the GOEV database and the non-redundant sequence database of GenBank. These sequences were used for construction of a phylogenetic tree.

To optimise sequence alignment and phylogenetic tree reconstruction, the following filtration steps were executed: (1) spurious detections were filtered out using hmmsearch against DVNP-HMM with the E-value cutoff of 1e-5, resulting in 905 sequences; (2) the remaining sequences were filtered based on length, using a formula involving the Interquartile range of the length of eight original DVNP sequences; (3) the 749 retained sequences were aligned using MAFFT; (4) sequences with gaps or misalignment for the most conserved motifs/sites were manually excluded. Then the Fasttree was used to reconstruct the phylogenetic tree. If an abnormally long branch was present in the phylogenetic tree, we removed the corresponding sequences from the FASTA file and repeated the multiple sequence alignment tree reconstruction steps. Finally, an approximately maximum likelihood phylogenetic tree was reconstructed based on the generated DVNP database, using the JTT + CAT model. Local support values ≥ 85 were shown by grey circles.

NLStradamus (http://www.moseslab.csb.utoronto.ca/NLStradamus/) was employed to identify the NLS of DVNPs by the cutoff value of 0.9[48]. This tool uses HMMs to predict NLS in proteins, which often contain specific residues[48].

## Structural prediction of DVNPs

AlphaFold2.0, as implemented in UCSF ChimeraX, was applied to predict protein structures of DVNP representatives[101]. Predicted models were visualised with UCSF ChimeraX. UCSF ChimeraX Matchmaker command was used for the structural alignment of the models[102]. Matchmaker considers residue similarity, secondary structure and gap penalties for model superposition.

## Core gene identification

To identify core genes shared by the group of flagellated-protist-infecting viruses of *Imitervirales*, whose names and accession numbers are listed below. The group of flagellated-protist-infecting viruses includes the following viruses: *Aureococcus anophagefferens* virus (AaV, KJ645900.1), (BsV, MF782455.1), *Chrysochromulina ericina* virus (CeV-01B, NC_028094.1), *Cafeteria roenbergensis* virus BV-PW1 (CroV, NC_014637.1), Organic Lake phycodnavirus 1 (OLPV-1, HQ704802.1), OLPV-2 (HQ704803.1), *Phaeocystis globosa* virus 16T (PgV-16T, NC_021312.1), *Tetraselmis* virus 1 (TetV1, KY322437.1) and *Prymnesium kappa* virus RF01 (PkV RF01, GCF_905367645.1).

The local protein database containing all the protein sequences of these viruses was constructed by blast+[103]. The all-to-all blast analysis of the local protein database was applied with cutoffs of E-value < 1e-5, sequence identity >35% and alignment coverage >60%. This analysis resulted in a large number of sequence hits with significant similarity[104]. The genes shared by all the viruses listed above are designated as 'core genes'[104]. Additionally, NCLDV genes were identified by hmmsearch against the NCLDV database with default settings[105]. To confirm functional similarity among the core genes, we also searched for conserved domains within the sequences. This ensures that each group of core genes contains a similar domain organization. The conserved domains were identified by hmmsearch (hmmer.org) against Pfam[94] and VOGDB databases, with the E-value cutoff of 1e-5. The identified core genes with their conserved domains are summarised in Supplementary Table 2.

### Repeat sequence analyses
Repeat sequences of HeV RF02 and PkV RF02 genomes were identified by repeat-match of the MUMmer package. All the repeats longer than 20 bp were extracted[106].

### Protein identification by LC-MS/MS
In-solution digestion and liquid chromatography coupled to tandem mass spectrometry (LC-MS/MS) analysis were used to identify all proteins present in purified PkV RF02 samples. Mass spectrometry analyses for protein identification were run for two independently purified PkV RF02 virus samples.

Each sample was diluted with lysis buffer (5% sodium dodecyl sulfate and 25 mM triethylammonium bicarbonate). Samples were reduced and alkylated by adding 5 mM tris(2-carboxyethyl) phosphine (TCEP) (pH 8.0) and 10 mM chloroacetamide (CAA) or methyl methanethiosulfonate (MMTS, Pierce), for 30 min at 60 °C. Protein digestion was performed in S-Trap columns (PROTIFI) following the manufacturer's instructions with minor changes[107]. Each sample was digested at 37 °C overnight using a trypsin:protein ratio of 1:15 and cleaned with a StageTip C18 or Oligo R3 polymeric reversed phase prior to LC-MS/MS analysis.

Peptide samples were analysed on a nano liquid chromatography system (Ultimate 3000 nano HPLC system, Thermo Fisher Scientific), coupled to an Orbitrap Exploris 240 mass spectrometer (Thermo Fisher Scientific). Approximately 500 ng of peptide sample was injected on a C18 PepMap trap column (5 µm, 100 µm I.D. × 2 cm, Thermo Scientific) at 10 µL/min. Equilibration was done in mobile phase A (0.1% formic acid (FA) in water) and the sample was switched on-line to a C18 PepMap Easy-spray analytical column (2 µm, 100 Å, 75 µm I.D. × 50 cm, Thermo Scientific). Peptides were eluted into the Easy-spray analytical column on a 60-min gradient from 2% to 95% mobile phase B (100% acetonitrile, 0.1% FA) at 45 °C and 250 nL min$^{-1}$.

Data acquisition was performed using a data-dependent top-20 method for protein identification, in full-scan positive mode, scanning 350–1200 m/z. Survey scans were acquired at a resolution of 60,000 at m/z 200, with a normalized automatic gain control (AGC) target (%) of 300 and a maximum injection time (IT) of 40 ms. The top-20 most intense ions from each precursor peptide spectra (MS1) were selected and fragmented via higher-energy collisional dissociation (HCD), by 32 collision energy. The resolution for HCD spectra was set to 45,000 at m/z 200, with an AGC target of 200 and a maximum ion IT of 120 ms. Isolation of precursors was performed with a window of 1 m/z, and an exclusion duration (s) of 10. Precursor ions with single, unassigned, or six and higher charge states from fragmentation selection were excluded.

MS1 and fragmented ionised peptide (MS2 or MS/MS) raw data were processed using Proteome Discoverer version 2.5.0.400 (Thermo Fisher Scientific). MS2 spectra were searched using Mascot Server v2.8.0 (Matrix Science, London, UK) and a target database built from sequences in the *Prymnesium kappa virus* PkV RF02 (PkV RF02), *Emiliania huxleyi* (GCF_000372725.1) and *Chrysochromulina tobinii* (GCA_001275005.1) proteomes. All searches were configured with dynamic modifications for

pyrrolidone from Q (−17.027 Da), oxidation of methionine residues (+15.9949 Da) and static modification as carbamidomethyl (+57.021 Da) on cysteine, monoisotopic masses and trypsin cleavage (maximum 2 missed cleavages). The peptide precursor mass tolerance was 10 ppm, and MS/MS tolerance was 0.02 Da. The false discovery rate for proteins, peptides and peptide spectral matches (PSMs) were kept at 1%. Only peptides with a significant individual ion score of at least 20 were considered. Protein normalised abundance was calculated by the abundance (intensity) of each peptide assigned to each protein and is shown in percentages in Table 2 and Supplementary Data 2. Percentages were calculated by dividing each protein abundance value by the sum of normalised abundances for all the proteins detected in the sample, including virus proteins and host proteins, as well as digestion enzymes introduced in the experiment.

Genome maps were visualised by Circos v0.69-9[108].

### Reporting summary
Further information on research design is available in the Nature Portfolio Reporting Summary linked to this article.

## Data availability
Genomic sequences of HeV RF02 and PkV RF02 have been deposited in GenBank with the accession numbers 'PV100843' and 'PV100844', respectively. The mass spectrometry proteomics data have been deposited to the ProteomeXchange Consortium via the PRIDE partner repository with the dataset identifier PXD060762 and https://doi.org/10.6019/PXD060762. Source data for Fig. 2 is available in Figshare with the identifier (https://doi.org/10.6084/m9.figshare.28381625).

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

## Acknowledgements

This work is supported by the RCN grants VirVar (Project no. 294363. Uncovering key players for regulation of phytoplankton function and structure: lessons to be learned from algal virus-haptophyte coexistence). We are grateful to Hilde Marie Kristiansen Stabell for her efficient technical support. We would like to thank Endy Spriet of Molecular Imaging Center (MIC) of University of Bergen for her assistance in electron microscopy. We thank Sergio Ciordia, Rosana Navajas and Fátima Santos at the proteomics facility of Centro Nacional de Biotecnología (CSIC) for expert LC-MS/MS assays. We acknowledge funding by grants from the Spanish State Research Agency, with co-funding from the European Regional Development Fund (PID2019-104098GB-I00/AEI/10.13039/501100011033 and PID2022-136456NB-I00/AEI/10.13039/501100011033), as well as CSIC (2023AEP082). The CSM group is a member of CSIC LifeHub (202120E47) and CSIC BCBHub. S.O-U holds a Spanish Ministry of Science, Innovation and Universities FPU predoctoral contract (FPU20/05148). G.N.C was a recipient of an EMBO short-term fellowship (STF-8085-2019).

## Author contributions

R.A.S. initiated the project. D.B. and J.K. sequenced and assembled the genomes. H.W. prepared and observed the samples for transmission electron microscopy (TEM). H.W. and M.R.L. took the samples. H.W., R.B-M. and H.D. analyzed the genomic sequences. H.W., S.O-U., G.N.C. and C.S.M. analyzed the proteome of PkV RF02 virions. H.W., L.M. and H.O. carried out the phylogenetic analysis of dinoflagellate/viral nucleoprotein homologues. R.S. predicted the SPs and transmembrane domains (TMs) of hypothetical proteins. H.W. wrote the original draft. H.W., R.A.S., S.O-U., G.N.C., C.S.M., L.M., H.O., R.S., M.R.L. and H.D. reviewed and edited the manuscript.

## Funding

## Competing interests

The authors declare no competing interests.
