## [Transparent Peer Review file · Communications Biology]

Haptophyte-infecting viruses change the genome condensing proteins of dinoflagellates

Corresponding Author: Professor Ruth-Anne Sandaa

This manuscript has been previously reviewed at another journal. This document only contains information relating to versions considered at Communications Biology.

Version 0:

Reviewer comments:

Reviewer #1

(Remarks to the Author)

The paper "Discovery from in-depth molecular analysis of giant viruses: Haptophyte-infecting viruses change the genome condensing proteins of dinoflagellates" by Wang et al reports on characterization of two giant viruses HeV RF02 and PkV RF02. The authors characterize the viruses in a number of ways including thin section electron microscopy of infected cells, whole genome sequencing and annotation, phylogenetic analysis and mass spectrometry proteomics to confirm proteins assembled into mature virions. Overall the work performed is solid and the data generated add to the growing body of knowledge for giant viruses. The electron microscopy is well done. In addition, I applaud the authors on using dual sequencing methods-Illumina and nanopore, both of which give a complete picture of the genome architecture and confidence in detecting long terminal repeats.

While the data are well done, the presentation of the manuscript is extremely difficult to follow and in most cases the authors conclusions are not well supported.

It appears most (if not all) of the authors are non-native English speakers. I would highly suggest they use a professional writing service. There are a ton of grammar issues, which is to be expected of ESL speakers. However, this is not the biggest problem with the writing. Simple grammar mistakes are easily fixed. The logic is often convoluted and in many cases the poor use of language obfuscates the meaning rendering many statements inaccurate or misleading. In addition the flow of the manuscript is very difficult to read and too terse leaving out many important details. The writing needs a complete overhaul before this work can be considered for publication in any journal.

Below I list some examples, but this is not extensive as there are many flaws--I cannot report on them all.

MAJOR ISSUES:

- Introduction: This section is very short and does not place the work in the context of the field. What major gaps in understanding are they aiming to address? What has been done in other giant virus families?
- Results: Many details are missing.
 - o I finally found the genome size buried deep in the materials and methods. This should be stated upfront as genome analysis is the majority of the paper. The size of these genomes place these giant viruses in the smaller end of the range of sizes for giants and this should be stated.
 - o What metric(s) were used for classifying the proteins into different categories or functionality? This is crucial to know. Blast results? What cutoffs? Some of this is in the methods section, but is incomplete and should be moved to the results to place the findings in context.
 - o Other details are missing. For example the authors state "PkV RF02 encoded the complete pathway while HeV RF02 encoded part of the pathway" What was missing in RF02?
 - o Many/most statements in this section are listed as fact, when they are really just a hypothesis. For example, the authors state "There are 10 and 11 proteases encoded by HeV RF02 and PkV RF02 respectively, functioning in protein degradation and disaggregation, ubiquitin recycling, and virion maturation [12-16]." This cannot be inferred by genome sequencing

alone. The genes may resemble proteases in other systems (such as in the references provided), but without expressing them and confirming this function, it remains purely speculative. This type of assertion of fact is found throughout the manuscript and is very misleading.

o In the proteome analysis the authors state “Ninety-one ORFs are of unknown function.” I take this to mean there are proteins present in the virions but the function of these gene products are not known. What are these ORFs? Table S4 only shows 14 called “hypothetical”. These 91 should be listed.

o No information is given in the “Phylogenetic analysis of DVNPs” section. As per the NLStradamus inputs—what cutoff was used? What is considered “homologous”?

- Results: The Alpha Fold predictions of the DVNPs doesn't add anything meaningful in the current presentation. Since Alpha Fold predictions are based on multiple sequence alignments, by definition of course a protein that is similar would have a similar fold. Not sure what this adds?
- Discussion: I am curious about the long viral terminal repeats and would like to know more about this. It is hard to discern from Figure 2A and B if these are flanking core genes, non-core genes, or are randomly spread throughout. If these are indeed hot spots for horizontal gene transfer this could be explored/explained further. Also, what is the connection to the jumbo phage? Are the authors suggesting some relationship between giant viruses and jumbo phages?
- Figures: Supplemental Figure S2 is very difficult to interpret as there is so much information squished so small and nothing can be read. In addition, the red/green color palate is not accessible for color blind folks.

MINOR ISSUES:

- Acronyms are also very difficult to follow in this paper. Typically an acronym would be defined early on and then used consistently throughout. Also, given the large number of acronyms, a table of abbreviations would help the reader tremendously.
- It is unfortunate that there were no page or line numbers as that would have facilitated the review.

Reviewer #2

(Remarks to the Author)

Wang et al used microscopic, phylogenetic and proteomic analysis to understand the replication process of two haptophyte-infecting giant viruses and mapped potential HGT in dinoflagellates of DVNPs.

In this manuscript, Wang et al used multiple tools suggest a path towards understanding the intricate biology of giant viruses. The manuscript brings a lot of interesting bioinformatic/in silico analysis that definitely enhances knowledge in the field. The putative infection cycle provided in the current text is astonishing. Even though we still need to have “wet-lab” data to prove the pathways suggested, this is one of a few times that authors dared to suggest a more applied biochemical/mechanistic data for giant viruses.

The manuscript was very clearly and well written so I have no suggestions on this end.

The work was performed very thoroughly and I only have minor observations, that are described below.

Lines 53-54: These results are too summarized. There is no introduction to the result. Why was it important to analyze the phylogeny of DNA polymerase B? Please add more information here.

General text suggestions:

- Line 227: what does it entail to have different predicted NLS? Please conclude the sentence.
- It would be interesting to have a little more explanation on the similarities/differences between histones and DVNPs. Also, how would this gene correlate with the histone-linked genes in other giant viruses? Are they similar? There has been some papers on the subject and would be interesting to explore a parallel here (some references: 10.1016/j.molcel.2022.10.020, 10.1016/j.molcel.2022.10.020, 10.1016/j.molcel.2022.10.020, 10.1016/j.molcel.2022.10.020)

Reviewer #3

(Remarks to the Author)

This research paper provides novel insights into the biology and evolution of giant viruses that infect marine haptophytes through a combination of advanced techniques such as electron microscopy, phylogenomics, and virion proteomics. Overall, this is a high-quality study that expands our understanding of giant virus diversity, evolution, and host interactions. By employing multiple techniques, the authors were able to gain a thorough understanding of the viruses' genetic makeup and how they interact with their hosts at various levels. For instance, the detailed functional annotation and metabolic pathway analyses of the viral genomes revealed the extensive genetic repertoire these viruses possess for manipulating their hosts, providing new insights into the evolutionary arms race between viruses and their hosts.

Another significant finding was the discovery of dinoflagellate/viral nucleoprotein (DVNP) homologs in the viruses and the phylogenetic analysis suggesting a viral origin for dinoflagellate DVNPs. This novel finding expands our understanding of the evolutionary history of these viruses and their hosts, as well as the mechanisms they use to interact with each other. While the study is well-written and logically organized, there are a few areas that could be improved upon. For instance, the

long inverted terminal repeats (ITRs) identified in the viruses are an interesting feature, but their potential role as hotspots for horizontal gene transfer could be expanded upon more. Additionally, the structural modeling of DVNPs and comparison of nuclear localization signals (NLS) between viral and dinoflagellate homologs are intriguing, but experimental validation would strengthen the hypothesis that dinoflagellates acquired the NLS separately after obtaining the DVNP from viruses. To further broaden the impact of this study, discussion of the ecological implications of these findings for marine ecosystems and biogeochemical cycles could be included.

Additionally, comparison to other giant viruses, beyond just the haptophyte-infecting ones, could provide additional evolutionary context. Finally, some additional details on the methods used, such as the criteria for identifying core genes, would aid reproducibility.

The authors provide novel insights into the biology and evolution of giant viruses that infect marine haptophytes. The comprehensive approach used to characterize these viruses and their hosts, combined with the novel findings regarding the origin of DVNPs and the role of viruses in shaping the evolution of their hosts, make this study a great contribution to the field.

Here are some minor grammatical and spelling errors that should be corrected to improve the paper:

1. Page 1, lines 25-27: " and further that" does not read like a segue that maintains proper sentence structure.
2. Line 44 and throughout the manuscript: The word "harbor" should be "harbour" for consistency with British English spelling used elsewhere in the manuscript (e.g., "neighbouring", "colour").
3. Page 2, line 66: "The morphological changes of infected cells clearly illustrate the cellular reorganization from cell growth to virus proliferation." - The word "of" should be replaced with "in" to make the sentence grammatically correct.
4. Page 4, line 136: "Except for the amino acid synthesis, quaternary ammonium transporter is encoded for input of nitrogen." - The word "Except" should be "In addition to" for clarity. Also, the phrase "is encoded for input of nitrogen" is ambiguous or perhaps overly terse.
5. Page 5, line 179: "Ninty-one ORFs are of unknown functions." - "Ninty-one" should be spelled as "Ninety-one".
6. Lines 363 and 383 "Approximately Maximum-likelihood phylogenetic tree was reconstructed" maximum does not need capitalization or hyphen and line 383 does not have a full sentence.

Version 1:

Reviewer comments:

Reviewer #1

(Remarks to the Author)

In general, the authors have responded well to the critiques. I appreciate the extra details and clarifications provided. In addition, softening some of the statements to show what is speculative versus what is known helps tremendously.

There are still a variety of grammar issues (some English, some scientific). Perhaps a copy editor will help make the readability better.

Reviewer #2

(Remarks to the Author)

The authors were able to address all the concerns I had from the original submitted manuscript.

Reviewer #3

(Remarks to the Author)

I am generally satisfied with the authors' responses to improve their manuscript. The work will be a strong contribution to an intriguing field.

Detected Typos and Minor Issues:

1. Page 2, Abstract: "arsenals that functions during the infection cycle"
Correction: "arsenals that function during the infection cycle."
2. Page 3, Introduction: "artificial environment of wastewater and terrestrial ecosystems"
Correction: "artificial environments of wastewater and terrestrial ecosystems."
3. Page 12, Phylogenetic analysis of DVNPs: "we expanded the PSI-BLAST search (10-iteration, E-value<1e-3)"
Correction: Should read "(10 iterations, E-value < 1e-3)" for consistency.
4. Page 15, Discussion: "This involve a novel nucleoprotein fibre..."
Correction: "This involves a novel nucleoprotein fibre..."
5. Page 19, Discussion: "genes involved in metabolisms of nucleotides, amino acids..."
Correction: "genes involved in the metabolism of nucleotides, amino acids..."

**Response to Reviewer #1:**

The paper “Discovery from in-depth molecular analysis of giant viruses: Haptophyte-infecting viruses
change the genome condensing proteins of dinoflagellates” by Wang et al reports on characterization
of two giant viruses HeV RF02 and PkV RF02. The authors characterize the viruses in a number of ways
including thin section electron microscopy of infected cells, whole genome sequencing and annotation,
phylogenetic analysis and mass spectrometry proteomics to confirm proteins assembled into mature
virions. Overall the work performed is solid and the data generated add to the growing body of
knowledge for giant viruses. The electron microscopy is well done. In addition, I applaud the authors
on using dual sequencing methods-Illumina and nanopore, both of which give a complete picture of
the genome architecture and confidence in detecting long terminal repeats.

While the data are well done, the presentation of the manuscript is extremely difficult to follow and
in most cases the authors conclusions are not well supported.

The logic is often convoluted and in many cases the poor use of language obfuscates the meaning
rendering many statements inaccurate or misleading. In addition the flow of the manuscript is very
difficult to read and too terse leaving out many important details. The writing needs a complete
overhaul before this work can be considered for publication in any journal.

Below I list some examples, but this is not extensive as there are many flaws--I cannot report on them
all.

**MAJOR ISSUES:**

- • **Introduction: This section is very short and does not place the work in the context of the field.**
**What major gaps in understanding are they aiming to address? What has been done in other giant**
**virus families?**

*We have followed the suggestion of R1 to expand the introduction to include the background*
*information needed to provide information about other giant virus families. We have included*
*information about giant viruses being widespread in the environment infecting a wide range of hosts,*
*including their morphological traits, genome contents and infection cycles. We have provided general*
*information about the group of viruses we are focusing on, giant viruses infecting haptophytes, which*
*is an ecologically important group as it infects one of the major contributors to marine primary*
*production in different oceans. We emphasize that the biology of giant viruses infecting haptophytes*

are largely unknown due to the lack of virus-host model systems which is the main knowledge gap we
are aiming to address in the study. The expanded introduction can be found in the manuscript in L31-
63.

The section of Introduction now reads as follows. The revisions are highlighted in blue.

Giant viruses belong to a group of viruses distinguished by their exceptionally large size,
complex structures and extensive genomic diversity (Schulz, Abergel et al. 2022). They
constitute the viral phylum *Nucleocytoviricota*, and infect a broad range of eukaryotic
hosts (Sun, Yang et al. 2020), including multicellular animal, and unicellular protists. Giant
viruses are highly abundant and diverse in aquatic environments (Monier, Claverie et al.
2008, Hingamp, Grimsley et al. 2013, Clerissi, Desdevises et al. 2015, Li, Hingamp et al.
2018, Mihara, Koyano et al. 2018, Li, Endo et al. 2019, Endo, Blanc-Mathieu et al. 2020),
as well as artificial environment of wastewater and terrestrial ecosystems (Schulz, Roux
et al. 2020). In the ocean, metagenome analysis has shown that the viruses belonging to
the *Algavirales* and *Imitervirales* are the most dominant and widespread families within
these phyla (Meng, Delmont et al. 2023).

The discovery and investigation of Mimivirus infecting amoeba *Acanthamoeba polyphaga*,
as well as its relatives (Colson, La Scola et al. 2017), has revealed extraordinary features
of morphology, genomics and infection cycles. The virions' diameter ranges from ~120 nm
(Derelle, Monier et al. 2015) to as long as 1 μm (Philippe, Legendre et al. 2013), exhibiting
diverse morphologies, most of which are icosahedral capsids decorated by complex
turreted structures and sometimes glycan modifications (Rodrigues, dos Santos Silva et al.
2015, Schulz, Abergel et al. 2022). The general steps in the infection cycle of giant viruses
start with membrane fusion and endocytosis to enter the host cell, followed by replication
and assembly in the viral factory in the cytoplasm of the host cell or in the host nucleus
(Schulz, Abergel et al. 2022). Mature virions are released after host cell lysis (Schulz,
Abergel et al. 2022). More intriguingly, these viruses carry a significant array of genes
involved in cellular life, encompassing DNA replication and recombination, transcription,
translation and posttranslational modifications (PTMs), as well as substantial energy
metabolism (Fischer, Allen et al. 2010, Santini, Jeudy et al. 2013, Moniruzzaman,
Martinez-Gutierrez et al. 2020, Blanc-Mathieu, Dahle et al. 2021, Schulz, Abergel et al.
2022). The cellular genes may participate in extensive functions that showcase their
extraordinary capability for manipulating their hosts and with potential roles in ecology
and global nutrient cycles that are validated by transcriptomic analysis (Thiriet-Rupert,

Carrier et al. 2016, Moniruzzaman, Gann et al. 2018, Ku, Sheyn et al. 2020) and
biochemical experiments *in vitro* (Rosenwasser, Mausz et al. 2014, Monier, Chambouvet
et al. 2017, York 2017).

We have only a few representatives of viruses in the *Imitervirales* in culture. These are
viruses infecting heterotrophic protists such as *Stramenopiles* (Fischer, Allen et al. 2010)
and *Kinetoplastida* (Deeg, Chow et al. 2018), and phototrophic/mixotrophic protists such
as Pelagophytes (genus *Aureococcus*, (Moniruzzaman, Gann et al. 2018)), Chlorophytes
(genera *Pyramimonas* and *Tetraselmis* (Sandaa, Heldal et al. 2001, Schvarcz and Steward
2018)), and haptophytes (the genera *Haptolina*, *Prymnesium* and *Phaeocystis* (Sandaa,
Saltvedt et al. 2022). The viruses infecting haptophytes are an ecologically important
group as their hosts are widespread in many oceans, contributing to 30-50% of total
chlorophyll *a* biomass (Liu, Probert et al. 2009). Notably, all the cultured haptophyte
viruses so far described are giant viruses. All except one, *Emiliana huxleyi* virus (Sandaa,
Saltvedt et al. 2022), belong to the viral order of *Imitervirales* (Aylward, Abrahao et al.
2023). Despite the ecological importance of haptophyte viruses, there are few studies
focusing on the details of morphological, genomic and proteomic changes during the
different steps in their infection.

By the use of genomic, phylogenetic and proteomic analyses, along with transmission
electron microscopy (TEM) observation, we have studied in detail the different steps in
the viral infection using two giant marine viruses, *Haptolina ericina* virus RF02 (HeV RF02)
and *Prymnesium kappa* virus RF02 (PkV RF02), and their hosts as model systems
(Johannessen, Bratbak et al. 2015). Our findings show that the viruses harbour a large
genome reservoir towards host manipulation that is utilized across the whole infection
cycle. Additionally, they hold a pair of the longest recorded inverted terminal repeats (ITRs)
that may serve as a hot spot of horizontal gene transfer (HGT). The two viral genomes
were found to encode nucleoprotein homologues (i.e. dinoflagellate/viral nucleoproteins,
DVNPs). Based on phylogenetic analyses across extensive viral metagenomes, combined
with structural prediction, we suggest a new hypothesis for the evolutionary history of
the nucleoproteins of dinoflagellates.

*We hope that the revised introduction has addressed the concerns you raised.*

• **Results: Many details are missing.**

**o I finally found the genome size buried deep in the materials and methods. This should be stated**
**upfront as genome analysis is the majority of the paper. The size of these genomes place these giant**
**viruses in the smaller end of the range of sizes for giants and this should be stated.**

*In the original version of the manuscript, we stated the genome sizes of HeV RF02 and PkV RF02 at L*
*96-97, that is, “HeV RF02 possesses a 582-kb genome encoding 545 predicted open reading frames*
*(ORFs), while PkV RF02 harbors a 583-kb genome encoding 635 predicted ORFs.”. And the exact sizes*
*of the two viral genomes were stated in the Methods at L463-467. Given your suggestion to place the*
*two viruses in this study in the smaller end of the range of sizes of giant viruses, the context is improved*
*as follows.*

L96-97: HeV RF02 possesses a 582,139-bp genome encoding 545 predicted open reading
frames (ORFs), while PkV RF02 harbors a 583,284-bp genome encoding 635 predicted
ORFs.

*In general, genome size of giant viruses ranges from 100kb-1.2Mb (Filee, Pouget et al. 2008).To our*
*knowledge, the genomes of the two viruses described in this study are more than twice the size of all*
*the giant viruses infecting prasinophytes, like Bathyococcus (BpV) Osterococcus (OtV), Micromonas*
*(PpV) (Finke, Winget et al. 2017) that have genomes of approximately 200 kb. The same goes for the*
*Chloroviruses that has a genome of approximately 315-370 kb (Yamada, Onimatsu et al. 2006).*
*Therefore, we do not prefer to evaluate the genome sizes of HeV RF02 and PkV RF02 in the manuscript.*

**o What metric(s) were used for classifying the proteins into different categories or functionality?**
**This is crucial to know. Blast results? What cutoffs? Some of this is in the methods section, but is**
**incomplete and should be moved to the results to place the findings in context.**

*To address the reviewer’s concerns, we rewrite the section of “Functional annotation and classification*
*of predicted ORFs” in Methods to include details of gene prediction, functional annotations and*
*categories, and cutoffs of BLAST and hmmsearch and so on. The revised context reads as follows.*

L468-499:

Functional annotation and classification of predicted ORFs

Open reading frames (ORFs) were predicted by GeneMarkS with the option ‘virus’
(Besemer, Lomsadze et al. 2001) in the assembled genome sequences of HeV RF02 and

PkV RF02, respectively. tRNAscan web server (<http://lowelab.ucsc.edu/tRNAscan-SE/>)
was applied to identify tRNAs using the default mode (Lowe and Chan 2016, Chan and
Lowe 2019). Functional annotation of each ORF is defined manually based on full-length
alignment of BLASTp and match of conserved domains described as follows. Firstly, amino
acid sequences of ORFs were searched against public databases, including Virus-Host DB
(Mihara, Nishimura et al. 2016), RefSeq (Pruitt, Tatusova et al. 2007), UniRef90 (Suzek,
Wang et al. 2015), and COG (Tatusov, Galperin et al. 2000) with an E-value cutoff of 1e-5.
The annotations were furtherly confirmed by conserved domains identified by
hmmsearch (hmmer.org) against databases of Pfam (Finn, Bateman et al. 2014), VOGDB
(Trgovec-Greif, Hellinger et al. 2024) (<https://vogdb.org/>; VOGDB release 212) as well
as InterPro (Paysan-Lafosse, Blum et al. 2023) with E-value cutoff of 1e-5. In ambiguous
cases, for example, when using hmmsearch to identify conserved domains in protein
sequences, we may obtain several different hits that meet the threshold, some of which
overlap. In such instances, we select the hits with the lowest E-value in the same region
in the protein sequences to minimize the overlap between identified domains. The
functional annotations were also manually checked through an integral online tool,
GenomeNet (<https://www.genome.jp/>; alignment quality, length comparison to
canonical genes, and links with KEGG orthology). Giant viruses shared a largely conserved
life cycle (Colson, La Scola et al. 2017, Schulz, Abergel et al. 2022, Talbert, Henikoff et al.
2023), including the steps of virus entry, chromatin remodeling, DNA replication, DNA
repair and recombination, transcription, translation, posttranslational modification,
vesicle trafficking and virus morphogenesis. To understand how viruses infect and
manipulate their hosts using their genetic reservoir, we classified the annotated genes
into functional groups corresponding to the steps of infection cycle mentioned above.
The classification is based on the functional annotation of ORFs, combined with
functional information inferred from the references affiliated with the conserved
domains in InterPro (Paysan-Lafosse, Blum et al. 2023) as well as the information
available in the EcoCyc Database (Keseler, Gama-Castro et al. 2021). To expand the
functional annotation of hypothetical proteins with unknown function, we predicted
signal peptides and transmembrane domains using SignalP6 (Teufel, Armenteros et al.
2022) and DeepTMHMM (Hallgren, Tsigos et al. 2022) using default settings. For some
types of putative enzymes may function in extensive biological processes, we searched
for information in specialized database to infer their functions. The functional
classification of glycosyltransferases (GTs) and proteases were defined by the databases

CAZy (Drula, Garron et al. 2022) and MEROPS (Rawlings, Waller et al. 2014) respectively.
The metabolic pathways were generated based on the information of EcoCyc Database
(Keseler, Gama-Castro et al. 2021).

*Following the reviewer's suggestion, we also include functional annotation method in the section of*
*"Results" briefly. The revised context can be found in the manuscript in L115-118. It now reads as*
*follows.*

L115-118: ORFs are functionally annotated by BLASTp (E-value<1e-5) and hmmsearch (E-
value<1e-5), as described in detail in Methods. The biological processes that the gene
may be involved in are inferred from the references affiliated by the conserved domains
in InterPro (Paysan-Lafosse, Blum et al. 2023), as well as the information available in the
EcoCyc Database (Keseler, Gama-Castro et al. 2021).

*The criteria of different categories are also mentions in the related paragraph. For example, at Line*
*124-127, we mentioned that "Five and fourteen ORFs were classified in the "Virus Entry" categories of*
*HeV RF02 and PkV RF02, respectively. Most of their protein products are membrane-associated*
*proteins with predicted ligand binding motifs, indicating their possible functions in "Virus entry". ORFs*
*with DNA binding domains are classified in the category of "Chromatin remodeling".*

**o Other details are missing. For example the authors state "PkV RF02 encoded the complete**
**pathway while HeV RF02 encoded part of the pathway" What was missing in RF02?**

*We have added the missing information at Line 136-138. The revised sentence is as follows.*

L136-138: PkV RF02 encode a complete set of genes potentially required for the BER
pathway, while HeV RF02 encode part of the pathway with the absence of the homologue
of UDG.

**o Many/most statements in this section are listed as fact, when they are really just a hypothesis.**
**For example, the authors state "There are 10 and 11 proteases encoded by HeV RF02 and PkV RF02**
**respectively, functioning in protein degradation and disaggregation, ubiquitin recycling, and virion**
**maturation [12-16]." This cannot be inferred by genome sequencing alone. The genes may resemble**
**proteases in other systems (such as in the references provided), but without expressing them and**

**confirming this function, it remains purely speculative. This type of assertion of fact is found**
**throughout the manuscript and is very misleading.**

*We agree with the reviewer's concern and have done necessary corrections in the manuscript to tone*
*down the claims to clarify that they are of hypothetical functions based on the sequence analysis*
*without biochemical evidence.*

One example is:

L160-162: There are 10 and 11 putative proteases encoded by HeV RF02 and PKV RF02,
respectively, which are inferred to function in protein degradation and disaggregation,
ubiquitin recycling and virion maturation.

*The revision is highlighted in blue in the above example. The similar corrections are all highlighted in*
*blue in the entire revised manuscript.*

**o In the proteome analysis the authors state “Ninety-one ORFs are of unknown function.” I take this**
**to mean there are proteins present in the virions but the function of these gene products are not**
**known. What are these ORFs? Table S4 only shows 14 called “hypothetical”. These 91 should be**
**listed.**

*The 91 hypothetical proteins are listed in Table S3.*

*We would like to clarify a potential misunderstanding that Table S4 is “Core genes of the flagellated-*
*protist-infecting viruses”.*

**o No information is given in the “Phylogenetic analysis of DVNPs” section. As per the NLStradamus**
**inputs—what cutoff was used? What is considered “homologous”?**

*To address the reviewer's concerns on the section of “Phylogenetic analysis of DVNPs”, we expanded*
*the context to include details of how to investigate the distribution of DVNPs homologues in marine*
*giant viruses, how to search for the DVNPs homologues and construct a comprehensive phylogenetic*
*tree, as well as the prediction of NLSs in DVNP homologues. The contents are revised both in Results*
*and Methods.*

*For the reviewers' specific question of NLStradamus inputs, we display the information in the revised*
*manuscript as follows.*

*L278-280 (Results):*

NLStradamus based on the hidden Markov model (HMM) (Nguyen Ba, Pogoutse et al.
2009) was applied to predict the NLS of the DVNPs (cutoff value of 0.9).

*L537-539 (Methods):*

NLStradamus (<http://www.moseslab.csb.utoronto.ca/NLStradamus/>) was employed to
identify the NLSs of DVNPs by the cutoff value of 0.9 (Nguyen Ba, Pogoutse et al. 2009).
This tool uses hidden Markov models (HMMs) to predict NLSs in proteins, which often
contain specific residues (Nguyen Ba, Pogoutse et al. 2009).

*When coming to the "homologous", the homology of DVNP sequences is determined by sequence*
*similarity, sequence alignment and phylogenetic tree constructed based on the sequence alignment as*
*described in "Results (L242-293)" and "Methods (L500-539)". The file of sequence alignment is*
*attached as a supplemental material.*

*They are displayed as follows. The revised contents are highlighted in blue.*

*Results (L242-293):*

Phylogenetic analysis of DVNPs

DVNP is a viral-derived protein acquired by dinoflagellate as the nucleoprotein replacing
histones, and represents a remarkable exception of eukaryotic chromatin proteins (Irwin,
Martin et al. 2018). The discovery of DVNP homologs in HeV RF02 and PkV RF02 prompted
a further investigation of the distribution of these homologs in related giant viruses. To
explore the distribution of DVNPs in the group of marine giant viruses, we **generated an**
**HMM file based on known DVNPs to search for their homologues** against the Global
Ocean Eukaryotic Viral (GOEV) database (Gaia, Meng et al. 2023). **PSI-BLAST was also**
**executed to detect the distant DVNP homologues in the database. This database contains**
**1817 genomes/viral metagenome-assembled genomes (vMAGs) that represents a**
**comprehensive repository for giant viruses in marine environments (Gaia, Meng et al.**
**2023). In total, 638 sequences were retrieved by the two searches, and 528 (29%) viral**
**genomes/vMAGs were found to encode DVNP homologues, primarily distributed in two**
**viral orders, *Imitervirales* (N = 419; 38%) and *Algavirales* (N = 98; 43%) (Fig. 4A). The**

distribution of DVNPs homologues in marine giant viruses is uneven. While a majority of
these viral genomes harbor a single copy of DVNP gene, a specific lineage within
*Imitervirales* encodes multiple copies (Fig. 4A). Within the order *Algavirales*,
prasinoviruses and phaeoviruses encode DVNP homologues, whereas chloroviruses do
not. Viruses with large genomes, such as mimiviruses and pandoraviruses infecting
amoeba, lack DVNP homologues. Very few detections of DVNP homologues were also
identified in viruses from genomes belonging *Asfuvirales*, *Pimascovirales*, and
*Pandoravirales*. However, given their sparse distribution these findings do not offer a
comprehensive representation of the respective taxa.

To investigate the origin and evolution of DVNPs, we expanded the PSI-BLAST search (10-
iteration, E-value<1e-3) to the NCBI non-redundant protein sequences database to
identify more DVNP homologues. This search resulted in 264 hits, most of which were
predominantly from genomes belonging dinoflagellates. The resultant sequences
combined with DVNP homologues identified in GOEV database were used for constructing
the phylogenetic tree. Additionally, the sequences were filtered and the sequence
alignment was refined for phylogenetic tree construction as described in Methods. The
phylogenetic tree of DVNPs sequences is predominantly separated into two major
subclades, i.e., *Imitervirales* and *Algavirales* (Fig. 4B). Notably, the cellular sequences
retrieved from the non-redundant database constituted a distinct monophyletic clade yet
nested within the *Imitervirales* clade, with the DVNPs of phaeoviruses (*Ectocarpus*
*siliculosus* virus (EsV) and *Feldmannia* sp. Virus (FsV)) in order of *Algavirales*.

These results showed that DVNP homologs are prevalent in marine giant viruses,
particularly within the order of *Imitervirales* and *Algavirales*. Our phylogenetic analysis
revealed that the cellular DVNPs are only found in dinoflagellates and points to their
potential origin from viruses of *Imitervirales*.

To investigate if the cellular DVNPs are able to enter the nucleus to replace the histones
we examined if these DVNPs contain the localization signal (NLS) that mediates the
transport of proteins synthesized in the cytoplasm into the nucleus. NLStradamus based
on hidden Markov model (Nguyen Ba, Pogoutse et al. 2009) was applied to predict the
NLSs of the DVNPs with cutoff value of 0.9. We found that the 672 out 718 DVNPs contain
NLSs. The multiple sequence alignment and predicted NLSs are displayed following the
rectangular phylogenetic tree of DVNPs (Fig. S2). The corresponding files are attached as
supplementary materials. Curiously, most viral homologues of DVNPs harbor NLSs at C-

terminus while DVNPs of dinoflagellate harbors NLSs close to the N-terminus. Both
termini contain highly variable regions in the DVNP sequences. Hence the NLSs are mostly
located in the highly variable regions. The crystal structure of any DVNP remains
unexplored due to the difficulty in protein expression of DVNPs of which protein products
form insoluble inclusion bodies in the expression cells (Gornik, Ford et al. 2012). Given
the challenges associated with DVNP expression, the structural prediction is essentially
significant as it provides additional structural information of DVNPs. To gain the insight
into the NLS localization in the 3-D structures, we predicted the 3-D structures of the two
DVNP protein sequences (ORF414 of PkV RF02 and DVNP.5) and highlight the predicted
NLSs (Fig. 4C). As a result, the localization of the predicted NLSs differed between the viral
and dinoflagellate DVNPs at the both levels of protein sequences and 3-D structures. This
implies that the predicted NLSs might be acquired from different sources.

*Methods (L500-539):*

**Phylogenetic analysis of DNA polymerases and DVNPs**

Protein sequences of DNA polymerases were collected from the Global Ocean Eukaryotic
Viral (GOEV) database (Gaia, Meng et al. 2023). These sequences were subsequently used
to construct the maximum-likelihood phylogenetic tree. -m TEST was used for the model
selection and pfam+F+I+G4 was chosen according to BIC (Bayesian Information Criterion).
The tree was rooted using the *Algavirales* clade.

We used DVNP sequences from HeV RF02 and PkV RF02, together with previously
reported viral DVNPs from *Ectocarpus siliculosus* virus (EsV) and *Feldmannia* sp. virus (FsV),
as well as reported cellular DVNPs (DVNP.5, 10, 12, 13) of dinoflagellate *Hematodinium*
sp. (Gornik, Ford et al. 2012). These sequences were used to execute multiple sequence
alignment by MAFFT – linsi. To obviate the influence of non-conservative regions, columns
with more than 60% gaps were excluded using Trimal. This generated a refined DVNP-
HMM file, which was then used for hmmsearch in the following steps. To investigate the
distribution of DVNP homologues in marine giant viruses, we used the DVNP-HMM file
generated in the previous step to perform hmmsearch (E-value<1e-3) against the Global
Ocean Eukaryotic Viral (GOEV) database. This database contains 1,817
genomes/metagenomic assembled genomes (MAGs), and represents a comprehensive
repository for giant viruses in marine environments (Gaia, Meng et al. 2023). Additionally,
we also used the DVNP sequence from dinoflagellate (AFY23224.1) as the seed for PSI-

BLAST (10-iteration, E-value<1e-3) against the GOEV database, due to its high average
identity with other DVNP sequences.

To explore the origin and evolution of DVNPs, the search for DVNP homologues was
expanded to a non-redundant sequence database of GenBank (Sayers, Bolton et al. 2022)
with the same query parameters and threshold as described above. In total, 910
homologue sequences of DVNP were obtained by hmmsearch and PSI-BLAST against the
GOEV database and the non-redundant sequence database of GenBank. These sequences
were used for construction of a phylogenetic tree.

To optimise sequence alignment and phylogenetic tree reconstruction, the following
filtration steps were executed: 1) spurious detections were filtered out using hmmsearch
against DVNP-HMM with the E-value cutoff of 1e-5, resulting in 905 sequences; 2) the
remaining sequences were filtered based on length, using a formula involving the
Interquartile Range (IQR) of the length of 8 original DVNP sequences; 3) the 749 retained
sequences were aligned using MAFFT; 4) sequences with gaps or misalignment for the
most conserved motifs/sites were manually excluded. Then the Fasttree was used to
reconstruct the phylogenetic tree. If an abnormally long branch was present in the
phylogenetic tree, we removed the corresponding sequences from the FASTA file and
repeated the multiple sequence alignment tree reconstruction steps. Finally, an
approximately maximum likelihood phylogenetic tree was reconstructed based on the
generated DVNP database, using the JTT + CAT model. Local support values ≥ 85 were
shown by grey circles.

NLStradamus (<http://www.moseslab.csb.utoronto.ca/NLStradamus/>) was employed to
identify the NLS of DVNPs by the cutoff value of 0.9 (Nguyen Ba, Pogoutse et al. 2009).
This tool uses hidden Markov models (HMMs) to predict NLS in proteins, which often
contain specific residues (Nguyen Ba, Pogoutse et al. 2009).

• **Results: The Alpha Fold predictions of the DVNPs doesn't add anything meaningful in the current**
**presentation. Since Alpha Fold predictions are based on multiple sequence alignments, by definition**
**of course a protein that is similar would have a similar fold. Not sure what this adds?**

*We thank the reviewer for the thoughtful feedback. We totally understand your concern regarding the*
*application of AlphaFold prediction of DVNPs that mainly based on multiple sequence alignment,*

*leading to the expectation that similar sequences generate similar folds. However, the AlphaFold*
*prediction is still of significance in our studies for the following reasons:*

*The AlphaFold prediction provides additional structural information of DVNPs. Although DVNPs of*
*dinoflagellates were discovered in 1972 (Rizzo and Nooden 1972) and have been characterized in detail*
*since 2012 (Gornik, Ford et al. 2012), the crystal structure of any DVNP remains unexplored due to the*
*difficulty in protein expression of DVNPs of which protein products form insoluble inclusion bodies in*
*the expression cells. Given the challenges associated with DVNP expression, the structural prediction*
*of them is essentially significant as it provides insights into their putative folds responding for DNA*
*binding. Furtherly, positions of potential NLSs can be highlighted on the three-dimensional map of the*
*interested DVNPs to display the varied localizations of NLSs among different sources of DVNPs (Fig.*
*4B), indicating different interaction mechanisms of transportation via nuclear pore.*

*To make a clearer context, we add the following sentences in the manuscript.*

L285-288: The crystal structure of any DVNP remains unexplored, due to the difficulty in
protein expression of DVNPs, of which protein products form insoluble inclusion bodies
in the expression cells (Gornik, Ford et al. 2012). Given the challenges associated with
DVNP expression, the structural prediction is essentially significant as it provides
additional structural information on DVNPs.

*We hope that the above explanation clarifies the important of AlphaFold predictions in our study.*

• **Discussion: I am curious about the long viral terminal repeats and would like to know more about**
**this. It is hard to discern from Figure 2A and B if these are flanking core genes, non-core genes, or**
**are randomly spread throughout. If these are indeed hot spots for horizontal gene transfer this could**
**be explored/explained further. Also, what is the connection to the jumbo phage? Are the authors**
**suggesting some relationship between giant viruses and jumbo phages?**

*We appreciate the reviewer's suggestions and have done the following adjustments to meet his*
*concern and suggestions.*

*We agree that the positional relationship between the ITRs and core genes could be more clearly*
*illustrated in the genomic map. To address this, we have highlighted both objects by red frames in*
*Figure 2 A and B (page 30 in revised manuscript, and L651-652) to emphasize their relative positions*
*on the genome maps. It can now be seen that the ITRs are located at the termini of the genomes, while*

*the majority of the core genes are concentrated in the central region, position inward and outside of*
*the ITRs.*

*We have expanded the discussion related to the roles of ITRs in the revised manuscript. The new version*
*provides a comprehensive overview with new references of the various functions of ITRs, ranging from*
*their involvement in genome topology and replication to their potential role in gene acquisition and*
*genome expansion. These potential functions have been investigated in several giant viruses, like*
*Mimivirus, Bodo saltans virus (BsV) and poxviruses, connecting the presence of ITRs with important*
*evolutionary processes of HGT and adaptation to host defenses.*

*The context of jumbo phage has been removed because it contains direct terminal repeats at both ends*
*of its genome, which is different from and weakly linked to ITRs investigated in this study.*

*Our revision can be found in the section of Discussion (L334-361) or see below directly.*

L334-361: Many giant viruses harbour ITRs ranging from hundreds of base pairs to 23 kb
that function in various ways (Strasser, Zhang et al. 1991, Raoult, Audic et al. 2004, Deeg,
Chow et al. 2018, Xia, Cheng et al. 2022). The 900-bp ITRs of Mimivirus may allow a
circular topology of the annealing (Raoult, Audic et al. 2004). In addition to the functions
of the short ITRs in the viral genome topology and replication, long ITRs of giant viruses
appear to be involved in the gene acquisition leading to genome expansion. All the long
ITRs mentioned, including those from HeV RF02, PkV RF02, poxviruses (McInnes, Damon
et al. 2023), *Bodo saltans virus* (BsV) (Deeg, Chow et al. 2018) and Megavirus Baoshan
(Me. Baoshan) (Xia, Cheng et al. 2022), include not only repeated elements, but also
hypothetical genes with putative essential functions. Some of these genes are expressed
and can be detected in both transcriptomic and proteomic analyses (Xia, Cheng et al.
2022). In the ITRs of BsV, the enrichment of ankyrin-repeat domains suggests a recent
sequence duplication expanding the viral genome (Deeg, Chow et al. 2018). Similarly, in
poxviruses, such as Mpox viruses causing global outbreaks in recent years, extensive ITR
expansion has been observed, involving gene duplication and loss (Brinkmann, Kohl et al.
2023). These genes may function in immune modulation, virulence and host adaptation
(Brinkmann, Kohl et al. 2023). In the model poxvirus vaccinia, studies in human cells have
shown that the recombination-mediated gene expansions can facilitate acquisition of
adaptive genetic elements of viruses (Elde, Child et al. 2012). These findings indicate that
gene acquisition in ITRs may play roles in the poxivirus adaptation of the host antiviral
defenses (Elde, Child et al. 2012, Brinkmann, Kohl et al. 2023). Additionally, extensive

lateral acquisition of genes from hosts and bacterial symbionts has been detected by
phylogenetic analysis in many NCLDV lineages (Filee, Pouget et al. 2008). The larger the
genome, the higher is the number of laterally acquired genes (Filee, Pouget et al. 2008).
Interestingly, although it remains unknown whether these viral genomes contain ITRs, all
of these laterally acquired genes show a strong tendency to be positioned at the
extremities of the viral genomes (Filee, Pouget et al. 2008). This pattern aligns with the
gene distribution in viral genomes of HeV RF02 and PkV RF02, where genes located in ITRs
are less conserved compared to those in the central region of the genome. Based on the
presence of enrichment of repeat sequences, and the functional genes validated by the
proteomic analyses, combined with the above analyses, it is tempting to suggest that ITRs
of HeV RF02 and PkV RF02 might be a potential region for horizontal gene transfer from
their hosts; and that gene acquisition by the long ITR may be the way of genetic innovation
or genome inflation in giant viruses.

• **Figures: Supplemental Figure S2 is very difficult to interpret as there is so much information**
**squished so small and nothing can be read. In addition, the red/green color palate is not accessible**
**for color blind folks.**

*We appreciate the reviewer's comments on the Supplemental Figure S2 to make it more readable.*

*Firstly, we would like to clarify that the dense information in the figure is intended to illustrate the*
*conserved and variable regions of DVNPs based on the sequence alignment, and the distribution of*
*NLSs across DVNPs from different sources by using different colour patterns. The figure is designed to*
*highlight the NLSs distribution rather than to convey specific sequence details. The optimized Figure S2*
*is inserted in page 38 of the revised manuscript.*

*For the readers' convenience to check multiple sequence alignment of DVNP homologues and the*
*prediction of NLSs of them, we attached the corresponding files as "Supplementary information 6".*

**MINOR ISSUES:**

• **Acronyms are also very difficult to follow in this paper. Typically, an acronym would be defined**
**early on and then used consistently throughout. Also, given the large number of acronyms, a table**
**of abbreviations would help the reader tremendously.**

We agree that the use of acronyms may be challenging to follow and have done the following changes.
 Firstly, we have reviewed the entire manuscript to ensure that each acronym is clearly defined upon its
 first occurrence and is used consistently throughout the text. Secondly, to further assist readers, we
 have added a table of abbreviations at the end of the manuscript, which lists all the acronyms used as
 Table S5. It has been included in the manuscript in page 43 and also displayed as follows.

Table S5. Acronyms

Acronym	Full name
AaV	Aureococcus anophagefferens virus
BsV	Bodo saltans virus
CeV-01B	Chrysochromulina ericina virus
CroV	Cafeteria roenbergensis virus
OLPV-1	Organic Lake phycodnavirus 1
OLPV-2	Organic Lake phycodnavirus 2
PgV-16T	Phaeocystis globosa virus 16T
TetV1	Tetraselmis virus 1
PkV RF01	Prymnesium kappa virus RF01
HeV RF02	Haptolina ericina virus RF02
PkV RF02	Prymnesium kappa virus RF02
EsV	Ectocarpus siliculosus virus
FsV	Feldmannia sp. Virus
DVNP	Dinoflagellate/viral nucleoprotein
PTMs	Posttranslational modifications
ITRs	Inverted terminal repeats
TEM	Transmission electron microscopy
HGT	Horizontal gene transfer
NCLDVs	Nucleocytoplasmic large DNA viruses
ORFs	Open reading frames
RFC	Replication factor C
PCNA	Proliferating cell nuclear antigen
BER	Base excision repair
MMR	Mismatch repair
ER	Endoplasmic reticulum
3'UTR	3'Untranslated region
eIF4E	Eukaryotic initiation factor 4E
UPR	Unfolded protein response
UDG	Uracil-DNA glycosylase
UvdE	UV damage endonuclease
HJR	Holliday junction resolvase
GTs	Glycosyltransferases
IQR	Interquartile Range
HMMs	Hidden Markov models
GOEV	Global Ocean Eukaryotic Viral database
vMAGs	Viral metagenome-assembled genomes

• It is unfortunate that there were no page or line numbers as that would have facilitated the review.

*We assumed that page and line numbers of the manuscript should be generated by the submission*
*system. If this is not the case, we are sorry that this was not included in our submitted manuscript. We*
*have included line number in the revised manuscript.*

**Response to Reviewer #2:**

Wang et al used microscopic, phylogenetic and proteomic analysis to understand the replication
process of two haptophyte-infecting giant viruses and mapped potential HGT in dinoflagellates of
DVNPs.

In this manuscript, Wang et al used multiple tools suggest a path towards understanding the intricate
biology of giant viruses. The manuscript brings a lot of interesting bioinformatic/in silico analysis that
definitely enhances knowledge in the field. The putative infection cycle provided in the current text in
astonishing. Even though we still need to have “wet-lab” data to prove the pathways suggested, this is
one of a few times that authors dared to suggest a more applied biochemical/mechanistic data for
giant viruses.

The manuscript was very clearly and well written so I have no suggestions on this end.

The work was performed very thoroughly and I only have minor observations, that are described
below.

**Lines 53-54: These results are too summarized. There is no introduction to the result. Why was it**
**important to analyze the phylogeny of DNA polymerase B? Please add more information here.**

*We thank the reviewer for the suggestions. To address the phylogenetic relationship of HeV RF02 and*
*PkV RF02 with other nucleocytoplasmic large DNA viruses (NCLDVs), we constructed the phylogenetic*
*tree of the DNA polymerase B of Nucleocytoviricota, a commonly used marker gene for phylogenetic.*
*The information has been added at L78-80.*

*The added information reads as follows.*

L78-80: To address the phylogenetic relationship of HeV RF02 and PkV RF02 with other
nucleocytoplasmic large DNA viruses (NCLDVs), we constructed the phylogenetic tree of
the DNA polymerase B of *Nucleocytoviricota*.

**General text suggestions:**

**- Line 227: what does it entail to have different predicted NLS? Please conclude the sentence.**

*The different predicted NLSs implies that they might be acquired or generated from different sources.*

*This information has been added at L293.*

*Now it reads as follows.*

L293: This implies that the predicted NLSs might be acquired from different sources.

**- It would be interesting to have a little more explanation on the similarities/differences between**
**histones and DVNPs. Also, how would this gene correlate with the histone-linked genes in other**
**giant viruses? Are they similar? There has been some papers on the subject and would be interesting**
**to explore a parallel here (some references: 10.1016/j.molcel.2022.10.020,**
**10.1016/j.molcel.2022.10.020, 10.1016/j.molcel.2022.10.020, 10.1016/j.molcel.2022.10.020)**

*We thank the reviewer for the suggestion on the comparison between viral histones and DVNPs to*
*broaden the impact of the discussion. Due to these comments, we have expanded the discussion on*
*histones and DVNPs among giant viruses. We have also added some new references to enrich the*
*content of the discussion. The expanded content can be found in the revised manuscript in L315-327*
*and displayed here.*

L315-327: Due to their large genome size, compacting genomes within the capsids of giant
viruses poses significant challenges. Some giant viruses utilise structural proteins to
condense their expansive genomes (Talbert, Henikoff et al. 2023). In addition to DVNPs,
various giant viruses compact their genomes using histone-like repeats. These viral
histones are found in marseilleviruses, iridoviruses, medusaviruses and some deep-
branching viral superclades (Irwin and Richards 2024). It has been proposed that these
viral histones represent evolutionary intermediates between archaeal and eukaryotic
nucleosomes (Liu, Bisio et al. 2021, Valencia-Sanchez, Abini-Agbomson et al. 2021, Bryson,

De Ioannes et al. 2022, Irwin and Richards 2024). Although both viral DVNPs and histones
are involved in the genome condensation, they exhibit no similarity at the amino acid
sequence level. Beyond condensation via DVNPs and histones, Mimivirus employs a
unique mechanism to compact its 1.2-Mb genome. This involve a novel nucleoprotein
fibre, which is subsequently encased within a 30-nm diameter helical protein shell (Villalta,
Schmitt et al. 2022). The essential proteins involved in Mimivirus genome condensation
remain unknown. These diverse genome-condensing mechanisms observed in giant
viruses suggest that their condensation strategies may have evolved independently.

**Response to Reviewer #3:**

This research paper provides novel insights into the biology and evolution of giant viruses that infect
marine haptophytes through a combination of advanced techniques such as electron microscopy,
phylogenomics, and virion proteomics. Overall, this is a high-quality study that expands our
understanding of giant virus diversity, evolution, and host interactions. By employing multiple
techniques, the authors were able to gain a thorough understanding of the viruses' genetic makeup
and how they interact with their hosts at various levels. For instance, the detailed functional
annotation and metabolic pathway analyses of the viral genomes revealed the extensive genetic
repertoire these viruses possess for manipulating their hosts, providing new insights into the
evolutionary arms race between viruses and their hosts.

Another significant finding was the discovery of dinoflagellate/viral nucleoprotein (DVNP) homologs
in the viruses and the phylogenetic analysis suggesting a viral origin for dinoflagellate DVNPs. This
novel finding expands our understanding of the evolutionary history of these viruses and their hosts,
as well as the mechanisms they use to interact with each other.

While the study is well-written and logically organized, there are a few areas that could be improved
upon. For instance, the long inverted terminal repeats (ITRs) identified in the viruses are an interesting
feature, but their potential role as hotspots for horizontal gene transfer could be expanded upon more.
Additionally, the structural modeling of DVNPs and comparison of nuclear localization signals (NLS)
between viral and dinoflagellate homologs are intriguing, but experimental validation would
strengthen the hypothesis that dinoflagellates acquired the NLS separately after obtaining the DVNP
from viruses.

To further broaden the impact of this study, discussion of the ecological implications of these findings
for marine ecosystems and biogeochemical cycles could be included.

Additionally, comparison to other giant viruses, beyond just the haptophyte-infecting ones, could
provide additional evolutionary context. Finally, some additional details on the methods used, such as
the criteria for identifying core genes, would aid reproducibility.

The authors provide novel insights into the biology and evolution of giant viruses that infect marine
haptophytes. The comprehensive approach used to characterize these viruses and their hosts,
combined with the novel findings regarding the origin of DVNPs and the role of viruses in shaping the
evolution of their hosts, make this study a great contribution to the field.

**Long inverted terminal repeats (ITRs)**

*We agree that the identification of the ITRs is an intriguing area for further exploration. We refer to*
*our comments to these suggestions above (page 13-14, 374-416 in this file) providing detailed*
*information of our response in the revised manuscript. The revised context can be found in the section*
*of Discussion of the revised manuscript in L334-361.*

**Experimental validation of different origins of NLSs in DVNPs**

*Regarding the reviewer's suggestion for experimental validation of different origins of NLSs in DVNPs,*
*we fully understand its importance to strengthen the hypothesis.*

*A possible experiment for validating the hypothesis that dinoflagellate acquire the NLSs of DVNPs from*
*the sources other than their viral origins, is to construct the recombinant expression vector of a series*
*of DVNPs, including wild-type viral DVNP, dinoflagellate DVNP and DVNPs with foreign NLSs at N and*
*C terminus as well as the dinoflagellate DVNP with or without its native NLS. These expression vectors*
*with reporter gene (e.g. GFP gene) could be transfected into a model cell line to allow the investigation*
*of cellular localization of these recombinant DVNPs with NLSs from different sources, to verify the*
*functionality of the NLSs. Further, the fusion vectors could be constructed that replace the NLS of viral*
*DVNP with the dinoflagellate DVNP NLS, and vice versa. The fusion vector could be transfected into the*
*proper cell line to detect the cellular localization of the fused NLS to test whether the NLS from viral*
*DVNPs can replace the NLS in dinoflagellate DVNPs, and vice versa. However, there are several*
*limitations of the above design, that complicates possible experiments.*

*First, the above experiments are designed based on the available cell lines. However, no cell lines exists*
*for dinoflagellate with genetic operation system. An option is to use yeast instead. Fortunately, a*
*pioneer study has revealed that the wild type of dinoflagellate DVNP can be localized in the nucleus by*

*its native NLS in yeast (Irwin, Martin et al. 2018). This suggests that yeast is an available model system*
*for the above experiments.*

*Second, the experiment should be designed based on the hypothesis that the differences in NLSs*
*between viral and dinoflagellate DVNPs lead to functional differences in their ability to direct DVNPs*
*to the nucleus. This implies that the NLSs must be intrinsically compatible with the nuclear import*
*machinery of their native hosts (Lu, Wu et al. 2021). Therefore, while yeast might be a suitable model*
*system to test NLS functionality, it might not accurately reflect their nuclear localization in*
*dinoflagellate and haptophyte.*

*Last, the previous study has shown that the wide type of dinoflagellate DVNP.5 and its fusion protein*
*with SV40 NLS can localized in the yeast nucleus (Irwin, Martin et al. 2018). This indicates that yeast*
*has a broad compatibility with different NLSs from dinoflagellate and mammal's virus SV40. The broad*
*compatibility observed in yeast is useful for demonstrating that an NLS is functional across different*
*species but cannot reflect the specificity that might present in more specialized systems like*
*dinoflagellates and hosts of Imitervirales.*

*Although we are unable to conduct experiments validating the hypothesis, we have expanded the*
*discussion to include additional literatures to support our hypothesis. The expanded discussion of*
*DVNPs can be found in the manuscript L315-327. It is also referred to the response to Reviewer2 on*
*page 17-18 in this file.*

*We have also modified our hypothesis as mentioned above L311-314. It now reads:*

L311-314: Hence the dinoflagellates may acquire DVNPs from viruses of *Imitervirales*.
Furthermore, their NLS directing the localisation in the nucleus might have been
generated from sources other than viruses belonging to *Imitervirales*, which could have
contributed to the replacement of histones.

**Discussion of ecological implications and evolutionary context**

*In response to the reviewer's suggestion to broaden the discussion to include the ecological*
*implications of our findings, we have added a section discussing the potential impact of HeV RF02 and*
*PkV RF02 on marine ecosystems and biogeochemical cycles. And we have added a discussion on the*
*evolutionary implications of our findings, particularly regarding the gene acquisition of cellular genes*
*by giant viruses combined with the comparison with ITRs in different giant viruses. This provides*
*additional evolutionary context and insights into the strategies of HGT and genome evolution in giant*
*viruses. We believe that this addition provides a more comprehensive understanding of the significance*

*of our finding in broader ecological context. The discussion of ITRs of giant viruses are referred to page*
*16-17, L467-509 in this file.*

*The other revised section of the manuscript is as follows.*

L413-427: The viruses in this study regulate haptophytes, an ecologically significant and
highly diverse group of marine algae that include both bloom-forming and non-bloom-
forming species (Sandaa, Saltvedt et al. 2022). The structure of haptophyte
communities in marine ecosystems is influenced not only by physico-chemical factors,
but also by viral infections (Sandaa, Saltvedt et al. 2022). Through host cell lysis, viruses
affect not only the abundance of haptophytes, but also indirectly influence
biogeochemical cycles cycle (Wilhelm and Suttle 1999) and the efficiency of the
biological carbon pump (Kaneko, Blanc-Mathieu et al. 2021).

Our study offers new insights into the intricate strategies that viruses employ to
manipulate host populations throughout the entire infection cycle, from host cell entry
to the release of new viral particles. Furthermore, we propose that ITRs might facilitate
HGT between giant viruses and their host. This hypothesis is supported by the
observation that laterally transferred genes are often located at extremities of the viral
genomes. A notable example of HGT identified in our study is the viral origin of
dinoflagellate DVNPs, which likely results from gene exchange between giant viruses and
hosts. These novel findings significantly enhance our understanding of the evolutionary
history of these giant viruses and their hosts, as well as the potential molecular
mechanisms underlying their interactions.

**The criteria of identification of core genes**

*We thank the reviewer for the suggestion regarding the reproducibility of core genes identifications.*
*In response, we have explained the procedure for identifying core genes in detail in the revised*
*manuscript (L546-565). The revised section is as follows.*

L546-565: Core gene identification

To identify core genes shared by the group of flagellated-protist-infecting viruses of
*Imitervirales*, whose names and accession numbers are listed below. The group of
flagellated-protist-infecting viruses includes the following viruses: *Aureococcus*
*anophagefferens* virus (AaV, KJ645900.1), *Bodo saltans* virus (BsV, MF782455.1),

*Chrysochromulina ericina* virus (CeV-01B, NC_028094.1), *Cafeteria roenbergensis* virus
BV-PW1 (CroV, NC_014637.1), Organic Lake phycodnavirus 1 (OLPV-1, HQ704802.1),
OLPV-2 (HQ704803.1), *Phaeocystis globosa* virus 16T (PgV-16T, NC_021312.1),
*Tetraselmis* virus 1 (TetV1, KY322437.1) and *Prymnesium kappa* virus RF01 (Pkv RF01,
GCF_905367645.1). Genomes of HeV RF02 and Pkv RF02 were attached.

The local protein database containing all the protein sequences of these viruses was
constructed by blast+ (Camacho, Coulouris et al. 2009). The all-to-all blast analysis of the
local protein database was applied with cutoffs of E-value<1e-5, sequence identity>35%
and alignment coverage>60%. This analysis resulted in a large number of sequence hits
with significant similarity (Huang, Jiao et al. 2021). The genes shared by all the viruses
listed above are designated as "core genes" (Huang, Jiao et al. 2021). Additionally, NCLDV
genes were identified by hmmsearch against the NCLDV database with default settings
(Aylward, Moniruzzaman et al. 2021). To confirm functional similarity among the core
genes, we also searched for conserved domains within the sequences. This ensures that
each group of core genes contains a similar domain organization. The conserved domains
were identified by hmmsearch (hmmer.org) against Pfam (Finn, Bateman et al. 2014) and
VOGDB (<https://vogdb.org/>) databases, with the E-value cutoff of 1e-5. The identified
core genes with their conserved domains are summarised in Table S4.

**Here are some minor grammatical and spelling errors that should be corrected to improve the**
**paper:**

**1. Page 1, lines 25-27: " and further that" does not read like a segue that maintains proper sentence**
**structure.**

*The sentence is improved as follows.*

L27-28: Furthermore, the analysis shows that the dinoflagellate homologs were possibly
acquired from viruses of the order *Imitervirales*.

**2. Line 44 and throughout the manuscript: The word "harbor" should be "harbour" for consistency**
**with British English spelling used elsewhere in the manuscript (e.g., "neighbouring", "colour").**

*This has been corrected.*

**3. Page 2, line 66: "The morphological changes of infected cells clearly illustrate the cellular**
**reorganization from cell growth to virus proliferation." - The word "of" should be replaced with "in"**
**to make the sentence grammatically correct.**

*Corrected.*

**4. Page 4, line 136: "Except for the amino acid synthesis, quaternary ammonium transporter is**
**encoded for input of nitrogen." - The word "Except" should be "In addition to" for clarity. Also, the**
**phrase "is encoded for input of nitrogen" is ambiguous or perhaps overly terse.**

*The whole sentence is refined as follows.*

L185-186: In addition to the amino acid synthesis, the homologue of quaternary
ammonium transporter might potentially be involved in nitrogen uptake of host cells in
the ocean.

**5. Page 5, line 179: "Ninty-one ORFs are of unknown functions." - "Ninty-one" should be spelled as**
**"Ninety-one".**

*Corrected.*

**6. Lines 363 and 383 "Approximately Maximum-likelihood phylogenetic tree was reconstructed"**
**maximum does not need capitalization or hyphen and line 383 does not have a full sentence.**

*We have removed the capitalization and hyphen and revised this section to ensure it forms a complete*
*sentence. The corrected sentence now reads:*

L534-535: Finally, an approximately maximum likelihood phylogenetic tree was
reconstructed using based on the generated DVNP database using the JTT + CAT model.

*New references are also added in the revised manuscript.*

**References**

Aylward, F. O., J. S. Abrahao, C. P. D. Brussaard, M. G. Fischer, M. Moniruzzaman, H.
Ogata and C. A. Suttle (2023). "Taxonomic update for giant viruses in the order
Imitervirales (phylum Nucleocytoviricota)." *Arch Virol* **168**(11): 283.

Aylward, F. O., M. Moniruzzaman, A. D. Ha and E. V. Koonin (2021). "A phylogenomic
framework for charting the diversity and evolution of giant viruses." *PLoS Biol* **19**(10):
e3001430.

Besemer, J., A. Lomsadze and M. Borodovsky (2001). "GeneMarkS: a self-training
method for prediction of gene starts in microbial genomes. Implications for finding
sequence motifs in regulatory regions." *Nucleic Acids Res* **29**(12): 2607-2618.

Blanc-Mathieu, R., H. Dahle, A. Hofgaard, D. Brandt, H. Ban, J. Kalinowski, H. Ogata and
R. A. Sandaa (2021). "A persistent giant algal virus, with a unique morphology, encodes
an unprecedented number of genes involved in energy metabolism." *J Virol* **95**(8).

Brinkmann, A., C. Kohl, K. Pape, D. Bourquain, A. Thurmer, J. Michel, L. Schaade and A.
Nitsche (2023). "Extensive ITR expansion of the 2022 Mpox virus genome through gene
duplication and gene loss." *Virus Genes* **59**(4): 532-540.

Bryson, T. D., P. De Ioannes, M. I. Valencia-Sanchez, J. G. Henikoff, P. B. Talbert, R. Lee,
B. La Scola, K. J. Armache and S. Henikoff (2022). "A giant virus genome is densely
packaged by stable nucleosomes within virions." *Mol Cell* **82**(23): 4458-4470 e4455.

Camacho, C., G. Coulouris, V. Avagyan, N. Ma, J. Papadopoulos, K. Bealer and T. L.
Madden (2009). "BLAST+: architecture and applications." *BMC Bioinformatics* **10**: 421.

Chan, P. P. and T. M. Lowe (2019). "tRNAscan-SE: Searching for tRNA Genes in Genomic
Sequences." *Methods Mol Biol* **1962**: 1-14.

Clerissi, C., Y. Desdevises, S. Romac, S. Audic, C. de Vargas, S. G. Acinas, R. Casotti, J.
Poulain, P. Wincker, P. Hingamp, H. Ogata and N. Grimsley (2015). "Deep sequencing of
amplified Prasinovirus and host green algal genes from an Indian Ocean transect
reveals interacting trophic dependencies and new genotypes." *Environ Microbiol Rep*
**7**(6): 979-989.

Colson, P., B. La Scola, A. Levasseur, G. Caetano-Anolles and D. Raoult (2017).
"Mimivirus: leading the way in the discovery of giant viruses of amoebae." *Nat Rev*
*Microbiol* **15**(4): 243-254.

Deeg, C. M., C. T. Chow and C. A. Suttle (2018). "The kinetoplastid-infecting Bodo
saltans virus (BsV), a window into the most abundant giant viruses in the sea." *Elife* **7**.

Derelle, E., A. Monier, R. Cooke, A. Z. Worden, N. H. Grimsley and H. Moreau (2015).
"Diversity of Viruses Infecting the Green Microalga *Ostreococcus lucimarinus*." *J Virol*
**89**(11): 5812-5821.

Drula, E., M. L. Garron, S. Dogan, V. Lombard, B. Henrissat and N. Terrapon (2022). "The
carbohydrate-active enzyme database: functions and literature." *Nucleic Acids Res*
**50**(D1): D571-D577.

Elde, N. C., S. J. Child, M. T. Eickbush, J. O. Kitzman, K. S. Rogers, J. Shendure, A. P.
Geballe and H. S. Malik (2012). "Poxviruses deploy genomic accordions to adapt rapidly
against host antiviral defenses." *Cell* **150**(4): 831-841.

Endo, H., R. Blanc-Mathieu, Y. Li, G. Salazar, N. Henry, K. Labadie, C. de Vargas, M. B.
Sullivan, C. Bowler, P. Wincker, L. Karp-Boss, S. Sunagawa and H. Ogata (2020).
"Biogeography of marine giant viruses reveals their interplay with eukaryotes and
ecological functions." *Nat Ecol Evol* **4**(12): 1639-1649.

Filee, J., N. Pouget and M. Chandler (2008). "Phylogenetic evidence for extensive lateral
acquisition of cellular genes by Nucleocytoplasmic large DNA viruses." *BMC Evol Biol* **8**:
320.

Finke, J. F., D. M. Winget, A. M. Chan and C. A. Suttle (2017). "Variation in the Genetic
Repertoire of Viruses Infecting *Micromonas pusilla* Reflects Horizontal Gene Transfer
and Links to Their Environmental Distribution." *Viruses* **9**(5).

Finn, R. D., A. Bateman, J. Clements, P. Coghill, R. Y. Eberhardt, S. R. Eddy, A. Heger, K.
Hetherington, L. Holm, J. Mistry, E. L. Sonnhammer, J. Tate and M. Punta (2014). "Pfam:
the protein families database." *Nucleic Acids Res* **42**(Database issue): D222-230.

Fischer, M. G., M. J. Allen, W. H. Wilson and C. A. Suttle (2010). "Giant virus with a
remarkable complement of genes infects marine zooplankton." *Proc Natl Acad Sci U S A*
**107**(45): 19508-19513.

Gaia, M., L. Meng, E. Pelletier, P. Forterre, C. Vanni, A. Fernandez-Guerra, O. Jaillon, P.
Wincker, H. Ogata, M. Krupovic and T. O. Delmont (2023). "Mirusviruses link
herpesviruses to giant viruses." *Nature* **616**(7958): 783-789.

Gornik, S. G., K. L. Ford, T. D. Mulhern, A. Bacic, G. I. McFadden and R. F. Waller (2012).
"Loss of nucleosomal DNA condensation coincides with appearance of a novel nuclear
protein in dinoflagellates." *Curr Biol* **22**(24): 2303-2312.

Hallgren, J., K. D. Tsirigos, M. D. Pedersen, J. J. Almagro Armenteros, P. Marcatili, H.
Nielsen, A. Krogh and O. Winther (2022). "DeepTMHMM predicts alpha and beta
transmembrane proteins using deep neural networks." *bioRxiv*:
2022.2004.2008.487609.

Hingamp, P., N. Grimsley, S. G. Acinas, C. Clerissi, L. Subirana, J. Poulain, I. Ferrera, H.
Sarmiento, E. Villar, G. Lima-Mendez, K. Faust, S. Sunagawa, J. M. Claverie, H. Moreau, Y.
Desdevises, P. Bork, J. Raes, C. de Vargas, E. Karsenti, S. Kandels-Lewis, O. Jaillon, F.
Not, S. Pesant, P. Wincker and H. Ogata (2013). "Exploring nucleo-cytoplasmic large
DNA viruses in Tara Oceans microbial metagenomes." *ISME J* **7**(9): 1678-1695.

Huang, X., N. Jiao and R. Zhang (2021). "The genomic content and context of auxiliary
metabolic genes in roseophages." *Environ Microbiol* **23**(7): 3743-3757.

Irwin, N. A. T., B. J. E. Martin, B. P. Young, M. J. G. Browne, A. Flaus, C. J. R. Loewen, P. J.
Keeling and L. J. Howe (2018). "Viral proteins as a potential driver of histone depletion in
dinoflagellates." *Nat Commun* **9**(1): 1535.

Irwin, N. A. T., B. J. E. Martin, B. P. Young, M. J. G. Browne, A. Flaus, C. J. R. Loewen, P. J.
Keeling and L. J. Howe (2018). "Viral proteins as a potential driver of histone depletion in
dinoflagellates." *Nature Communications* **9**.

Irwin, N. A. T. and T. A. Richards (2024). "Self-assembling viral histones are evolutionary
intermediates between archaeal and eukaryotic nucleosomes." *Nat Microbiol*.

Johannessen, T. V., G. Bratbak, A. Larsen, H. Ogata, E. S. Egge, B. Edvardsen, W. Eikrem
and R. A. Sandaa (2015). "Characterisation of three novel giant viruses reveals huge
diversity among viruses infecting Prymnesiales (Haptophyta)." *Virology* **476**: 180-188.

Kaneko, H., R. Blanc-Mathieu, H. Endo, S. Chaffron, T. O. Delmont, M. Gaia, N. Henry, R.
Hernandez-Velazquez, C. H. Nguyen, H. Mamitsuka, P. Forterre, O. Jaillon, C. de Vargas,
762 M. B. Sullivan, C. A. Suttle, L. Guidi and H. Ogata (2021). "Eukaryotic virus composition
can predict the efficiency of carbon export in the global ocean." *iScience* **24**(1): 102002.

Keseler, I. M., S. Gama-Castro, A. Mackie, R. Billington, C. Bonavides-Martinez, R.
Caspi, A. Kothari, M. Krummenacker, P. E. Midford, L. Muniz-Rascado, W. K. Ong, S.
Paley, A. Santos-Zavaleta, P. Subhraveti, V. H. Tierrafria, A. J. Wolfe, J. Collado-Vides, I. T.
Paulsen and P. D. Karp (2021). "The EcoCyc Database in 2021." *Front Microbiol* **12**:
711077.

Ku, C., U. Sheyn, A. Sebe-Pedros, S. Ben-Dor, D. Schatz, A. Tanay, S. Rosenwasser and
770 A. Vardi (2020). "A single-cell view on alga-virus interactions reveals sequential
transcriptional programs and infection states." *Science Advances* **6**(21).

Li, Y., H. Endo, Y. Gotoh, H. Watai, N. Ogawa, R. Blanc-Mathieu, T. Yoshida and H. Ogata
(2019). "The Earth Is Small for "Leviathans": Long Distance Dispersal of Giant Viruses
across Aquatic Environments." *Microbes Environ* **34**(3): 334-339.

Li, Y., P. Hingamp, H. Watai, H. Endo, T. Yoshida and H. Ogata (2018). "Degenerate PCR
Primers to Reveal the Diversity of Giant Viruses in Coastal Waters." *Viruses* **10**(9).

Liu, H., I. Probert, J. Uitz, H. Claustre, S. Aris-Brosou, M. Frada, F. Not and C. de Vargas
(2009). "Extreme diversity in noncalcifying haptophytes explains a major pigment
paradox in open oceans." *Proc Natl Acad Sci U S A* **106**(31): 12803-12808.

Liu, Y., H. Bisio, C. M. Toner, S. Jeudy, N. Philippe, K. Zhou, S. Bowerman, A. White, G.
Edwards, C. Abergel and K. Luger (2021). "Virus-encoded histone doublets are essential
and form nucleosome-like structures." *Cell*.

Lowe, T. M. and P. P. Chan (2016). "tRNAscan-SE On-line: integrating search and context
for analysis of transfer RNA genes." *Nucleic Acids Res* **44**(W1): W54-57.

Lu, J., T. Wu, B. Zhang, S. Liu, W. Song, J. Qiao and H. Ruan (2021). "Types of nuclear
localization signals and mechanisms of protein import into the nucleus." *Cell Commun*
*Signal* **19**(1): 60.

McInnes, C. J., I. K. Damon, G. L. Smith, G. McFadden, S. N. Isaacs, R. L. Roper, D. H.
Evans, C. R. Damaso, O. Carulei, L. M. Wise and E. J. Lefkowitz (2023). "ICTV Virus
Taxonomy Profile: Poxviridae 2023." *J Gen Virol* **104**(5).

Meng, L., T. O. Delmont, M. Gaia, E. Pelletier, A. Fernandez-Guerra, S. Chaffron, R. Y.
Neches, J. Wu, H. Kaneko, H. Endo and H. Ogata (2023). "Genomic adaptation of giant
viruses in polar oceans." *Nat Commun* **14**(1): 6233.

Mihara, T., H. Koyano, P. Hingamp, N. Grimsley, S. Goto and H. Ogata (2018). "Taxon
Richness of "Megaviridae" Exceeds those of Bacteria and Archaea in the Ocean."
*Microbes Environ* **33**(2): 162-171.

Mihara, T., Y. Nishimura, Y. Shimizu, H. Nishiyama, G. Yoshikawa, H. Uehara, P.
Hingamp, S. Goto and H. Ogata (2016). "Linking Virus Genomes with Host Taxonomy."
*Viruses* **8**(3): 66.

Monier, A., A. Chambouvet, D. S. Milner, V. Attah, R. Terrado, C. Lovejoy, H. Moreau, A. E.
Santoro, E. Derelle and T. A. Richards (2017). "Host-derived viral transporter protein for
nitrogen uptake in infected marine phytoplankton." *Proc Natl Acad Sci U S A* **114**(36):
E7489-E7498.

Monier, A., J. M. Claverie and H. Ogata (2008). "Taxonomic distribution of large DNA
viruses in the sea." *Genome Biol* **9**(7): R106.

Moniruzzaman, M., E. R. Gann and S. W. Wilhelm (2018). "Infection by a Giant Virus
(AaV) Induces Widespread Physiological Reprogramming in *Aureococcus*
*anophagefferens* CCMP1984 - A Harmful Bloom Algae." *Front Microbiol* **9**: 752.

Moniruzzaman, M., C. A. Martinez-Gutierrez, A. R. Weinheimer and F. O. Aylward (2020).
"Dynamic genome evolution and complex virocell metabolism of globally-distributed
giant viruses." *Nat Commun* **11**(1): 1710.

Nguyen Ba, A. N., A. Pogoutse, N. Provar and A. M. Moses (2009). "NLStradamus: a
simple Hidden Markov Model for nuclear localization signal prediction." *BMC*
*Bioinformatics* **10**: 202.

Paysan-Lafosse, T., M. Blum, S. Chuguransky, T. Grego, B. L. Pinto, G. A. Salazar, M. L.
Bileschi, P. Bork, A. Bridge, L. Colwell, J. Gough, D. H. Haft, I. Letunic, A. Marchler-Bauer,
H. Mi, D. A. Natale, C. A. Orengo, A. P. Pandurangan, C. Rivoire, C. J. A. Sigrist, I. Sillitoe,
818 N. Thanki, P. D. Thomas, S. C. E. Tosatto, C. H. Wu and A. Bateman (2023). "InterPro in
2022." *Nucleic Acids Res* **51**(D1): D418-D427.

Philippe, N., M. Legendre, G. Doutre, Y. Coute, O. Poirot, M. Lescot, D. Arslan, V. Seltzer,
821 L. Bertaux, C. Bruley, J. Garin, J. M. Claverie and C. Abergel (2013). "Pandoraviruses:
amoeba viruses with genomes up to 2.5 Mb reaching that of parasitic eukaryotes."
*Science* **341**(6143): 281-286.

Pruitt, K. D., T. Tatusova and D. R. Maglott (2007). "NCBI reference sequences (RefSeq):
a curated non-redundant sequence database of genomes, transcripts and proteins."
*Nucleic Acids Res* **35**(Database issue): D61-65.

Raoult, D., S. Audic, C. Robert, C. Abergel, P. Renesto, H. Ogata, B. La Scola, M. Suzan
and J. M. Claverie (2004). "The 1.2-megabase genome sequence of Mimivirus." *Science*
**306**(5700): 1344-1350.

Rawlings, N. D., M. Waller, A. J. Barrett and A. Bateman (2014). "MEROPS: the database
of proteolytic enzymes, their substrates and inhibitors." *Nucleic Acids Res* **42**(Database
issue): D503-509.

Rizzo, P. J. and L. D. Nooden (1972). "Chromosomal proteins in the dinoflagellate alga
*Gyrodinium cohnii*." *Science* **176**(4036): 796-797.

Rodrigues, R. A., L. K. dos Santos Silva, F. P. Dornas, D. B. de Oliveira, T. F. Magalhaes, D.
836 A. Santos, A. O. Costa, L. de Macedo Farias, P. P. Magalhaes, C. A. Bonjardim, E. G.
Kroon, B. La Scola, J. R. Cortines and J. S. Abrahao (2015). "Mimivirus Fibrils Are
Important for Viral Attachment to the Microbial World by a Diverse Glycoside Interaction
Repertoire." *J Virol* **89**(23): 11812-11819.

Rosenwasser, S., M. A. Mausz, D. Schatz, U. Sheyn, S. Malitsky, A. Aharoni, E.
Weinstock, O. Tzfadia, S. Ben-Dor, E. Feldmesser, G. Pohnert and A. Vardi (2014).
"Rewiring Host Lipid Metabolism by Large Viruses Determines the Fate of *Emiliania*
*huxleyi*, a Bloom-Forming Alga in the Ocean " *The Plant Cell* **26**(6): 2689-2707.

Sandaa, R.-A., M. R. Saltvedt, H. Dahle, H. Wang, S. Vage, R. Blanc-Mathieu, I. H. Steen,
845 N. Grimsley, B. Edvardsen, H. Ogata and J. Lawrence (2022). "Adaptive evolution of
846 viruses infecting marine microalgae (haptophytes), from acute infections to stable
coexistence." *Biological Reviews* **97**(1): 179-194.

Sandaa, R. A., M. Heldal, T. Castberg, R. Thyraug and G. Bratbak (2001). "Isolation and
characterization of two viruses with large genome size infecting *Chrysochromulina*
*ericina* (Prymnesiophyceae) and *Pyramimonas orientalis* (Prasinophyceae)." *Virology*
**290**(2): 272-280.

Sandaa, R. A., M. R. Saltvedt, H. Dahle, H. Wang, S. Vage, R. Blanc-Mathieu, I. H. Steen,
853 N. Grimsley, B. Edvardsen, H. Ogata and J. Lawrence (2022). "Adaptive evolution of
854 viruses infecting marine microalgae (haptophytes), from acute infections to stable
coexistence." *Biol Rev Camb Philos Soc* **97**(1): 179-194.

Santini, S., S. Jeudy, J. Bartoli, O. Poirot, M. Lescot, C. Abergel, V. Barbe, K. E.
Wommack, A. A. Noordeloos, C. P. Brussaard and J. M. Claverie (2013). "Genome of
*Phaeocystis globosa* virus PgV-16T highlights the common ancestry of the largest known
DNA viruses infecting eukaryotes." *Proc Natl Acad Sci U S A* **110**(26): 10800-10805.

Sayers, E. W., E. E. Bolton, J. R. Brister, K. Canese, J. Chan, D. C. Comeau, R. Connor, K.
Funk, C. Kelly, S. Kim, T. Madej, A. Marchler-Bauer, C. Lanczycki, S. Lathrop, Z. Lu, F.
Thibaud-Nissen, T. Murphy, L. Phan, Y. Skripchenko, T. Tse, J. Wang, R. Williams, B. W.
Trawick, K. D. Pruitt and S. T. Sherry (2022). "Database resources of the national center
for biotechnology information." *Nucleic Acids Res* **50**(D1): D20-D26.

Schulz, F., C. Abergel and T. Woyke (2022). "Giant virus biology and diversity in the era of
genome-resolved metagenomics." *Nat Rev Microbiol* **20**(12): 721-736.

Schulz, F., S. Roux, D. Paez-Espino, S. Jungbluth, D. A. Walsh, V. J. Denef, K. D.
McMahon, K. T. Konstantinidis, E. A. Elloe-Fadrosh, N. C. Kyrpides and T. Woyke (2020).
"Giant virus diversity and host interactions through global metagenomics." *Nature*
**578**(7795): 432-+.

Schvarcz, C. R. and G. F. Steward (2018). "A giant virus infecting green algae encodes
key fermentation genes." *Virology* **518**: 423-433.

Strasser, P., Y. P. Zhang, J. Rohozinski and J. L. Van Etten (1991). "The termini of the
chlorella virus PBCV-1 genome are identical 2.2-kbp inverted repeats." *Virology* **180**(2):
763-769.

Sun, T. W., C. L. Yang, T. T. Kao, T. H. Wang, M. W. Lai and C. Ku (2020). "Host Range and
Coding Potential of Eukaryotic Giant Viruses." *Viruses* **12**(11).

Suzek, B. E., Y. Wang, H. Huang, P. B. McGarvey, C. H. Wu and C. UniProt (2015). "UniRef
clusters: a comprehensive and scalable alternative for improving sequence similarity
searches." *Bioinformatics* **31**(6): 926-932.

Talbert, P. B., S. Henikoff and K. J. Armache (2023). "Giant variations in giant virus
genome packaging." *Trends Biochem Sci* **48**(12): 1071-1082.

Tatusov, R. L., M. Y. Galperin, D. A. Natale and E. V. Koonin (2000). "The COG database: a
tool for genome-scale analysis of protein functions and evolution." *Nucleic Acids Res*
**28**(1): 33-36.

Teufel, F., J. J. A. Armenteros, A. R. Johansen, M. H. Gislason, S. I. Pihl, K. D. Tsirigos, O.
Winther, S. Brunak, G. von Heijne and H. Nielsen (2022). "SignalP 6.0 predicts all five
types of signal peptides using protein language models." *Nature Biotechnology* **40**(7):
1023-+.

Thiriet-Rupert, S., G. Carrier, B. Chenais, C. Trottier, G. Bougaran, J. P. Cadoret, B.
Schoefs and B. Saint-Jean (2016). "Transcription factors in microalgae: genome-wide
prediction and comparative analysis." *BMC Genomics* **17**: 282.

Trgovec-Greif, L., H. J. Hellinger, J. Mainguy, A. Pfundner, D. Frishman, M. Kiening, N. S.
Webster, P. W. Laffy, M. Feichtinger and T. Rattei (2024). "VOGDB-Database of Virus
Orthologous Groups." *Viruses* **16**(8).

Valencia-Sanchez, M. I., S. Abini-Agbomson, M. Wang, R. Lee, N. Vasilyev, J. Zhang, P.
De Ioannes, B. La Scola, P. Talbert, S. Henikoff, E. Nudler, A. Erives and K. J. Armache
(2021). "The structure of a virus-encoded nucleosome." *Nat Struct Mol Biol* **28**(5): 413-
417.

Villalta, A., A. Schmitt, L. F. Estrozi, E. R. J. Quemini, J. M. Alempic, A. Lartigue, V. Prazak,
901 L. Belmudes, D. Vasishtan, A. M. G. Colmant, F. A. Honore, Y. Coute, K. Grunewald and
902 C. Abergel (2022). "The giant mimivirus 1.2 Mb genome is elegantly organized into a 30-
903 nm diameter helical protein shield." *Elife* **11**.

Wilhelm, S. W. and C. A. Suttle (1999). "Viruses and Nutrient Cycles in the Sea: Viruses
play critical roles in the structure and function of aquatic food webs." *BioScience*
**49**(10): 781-788.

Xia, Y., H. Cheng and J. Zhong (2022). "Hybrid Sequencing Resolved Inverted Terminal
Repeats in the Genome of Megavirus Baoshan." *Front Microbiol* **13**: 831659.

Yamada, T., H. Onimatsu and J. L. Van Etten (2006). "Chlorella viruses." *Adv Virus Res*
**66**: 293-336.

York, A. (2017). "Marine microbiology: Algal virus boosts nitrogen uptake in the ocean."
*Nat Rev Microbiol* **15**(10): 573.

REVIEWERS' COMMENTS:

Reviewer #1 (Remarks to the Author):

In general, the authors have responded well to the critiques. I appreciate the extra details and clarifications provided. In addition, softening some of the statements to show what is speculative versus what is known helps tremendously.

There are still a variety of grammar issues (some English, some scientific). Perhaps a copy editor will help make the readability better.

We thank for R1's advice. We have used editing service before submission.

Reviewer #2 (Remarks to the Author):

The authors were able to address all the concerns I had from the original submitted manuscript.

Reviewer #3 (Remarks to the Author):

I am generally satisfied with the authors' responses to improve their manuscript. The work will be a strong contribution to an intriguing field.

Detected Typos and Minor Issues:

1. Page 2, Abstract: "arsenals that functions during the infection cycle"

Correction: "arsenals that function during the infection cycle."

2. Page 3, Introduction: "artificial environment of wastewater and terrestrial ecosystems"

Correction: "artificial environments of wastewater and terrestrial ecosystems."

3. Page 12, Phylogenetic analysis of DVNPs: "we expanded the PSI-BLAST search (10-iteration, E-value<1e-3)"

Correction: Should read "(10 iterations, E-value < 1e-3)" for consistency.

4. Page 15, Discussion: "This involve a novel nucleoprotein fibre..."

Correction: "This involves a novel nucleoprotein fibre..."

5. Page 19, Discussion: "genes involved in metabolisms of nucleotides, amino acids..."

Correction: "genes involved in the metabolism of nucleotides, amino acids..."

We thank reviewer's feedback and effort to improve our manuscript. The typos mentioned above are corrected.

It should be noted that the sentence "genes involved in metabolisms of nucleotides, amino acids..." is not in Discussion but in the section of Results and Table 1. We have corrected them as suggested.